## PROCEEDINGS A

flares, coronal mass ejections, magnetic fields, filaments/prominences, solar energetic particles, magnetic reconnection

**Author for correspondence:**
Yuandeng Shen
e-mail: ydshen@ynao.ac.cn

# Observation and modelling of solar jets

## Yuandeng Shen

Yunnan Observatories, Chinese Academy of Sciences, Kunming 650216, People's Republic of China

 YS, 0000-0001-9493-4418

The solar atmosphere is full of complicated transients manifesting the reconfiguration of the solar magnetic field and plasma. Solar jets represent collimated, beam-like plasma ejections; they are ubiquitous in the solar atmosphere and important for our understanding of solar activities at different scales, the magnetic reconnection process, particle acceleration, coronal heating, solar wind acceleration, as well as other related phenomena. Recent high-spatio-temporal-resolution, wide-temperature coverage and spectroscopic and stereoscopic observations taken by ground-based and space-borne solar telescopes have revealed many valuable new clues to restrict the development of theoretical models. This review aims at providing the reader with the main observational characteristics of solar jets, physical interpretations and models, as well as unsolved outstanding questions in future studies.

## 1. Introduction

The dynamic solar atmosphere hosts many jetting phenomena that manifest as collimated plasma beams with a width ranging from several hundred kilometres to a few times $10^5$ kilometres [1–5]; they are frequently accompanied by micro-flares, photospheric magnetic flux cancellations and type III radio bursts, and can occur in all types of solar regions including active regions, coronal holes and quiet-Sun regions. Since these jetting activities continuously supply mass and energy into the upper atmosphere, they are thought to be one of the most important sources for heating coronal plasma and accelerating solar wind [1,6–9].

This review mainly focuses on bigger solar jets, including surges, coronal jets and macro-spicules. Although these jet activities are observed at different scales and temperature ranges, they can be viewed as the

same type of solar jets owing to their similar observational characteristics and generation mechanism, i.e. magnetic reconnection-dominated jet-like activities with an inverted-Y structure. For smaller, lower-energy jet-like activities such as spicules and dynamic fibrils, their generation mechanisms are still open questions. Previous studies have suggested that spicules and dynamic fibrils are possibly launched by upward-propagating shocked pressure-driven waves leaking from the photosphere [10,11]. However, some recent studies have indicated that a portion of spicules are possibly produced by the same mechanism as bigger jets, since they also showed an inverted-Y shape and are associated with flux cancellations [9,12,13]. Hence, the generation of these small jet-like activities needs further investigation; however, the present review will not introduce them in detail. Readers who are interested in this topic can refer to several previous reviews [14–16].

The observation of solar jets dates back to the 1940s; they were dubbed surges in history [17]. At the very beginning, surges were found to be associated with micro-flares or sudden brightenings near their footpoints [18]. Later, observations suggested that surges were dominated by local magnetic fields [19]; they move along magnetic field lines and tend to occur above satellite sunspots or in regions of evolving magnetic features [20,21]. Before the 1990s, observations were mainly taken by small-aperture ground-based Hα telescopes and a few low-resolution space instruments such as *Skylab* (1991–2001) and *SMM* (1980–1989). Solar jets have been studied intensively since the 1990s owing to the launch of a series of space telescopes, including the *Yohkoh* satellite [22], the *Solar and Heliospheric Observatory* [23] (*SOHO*; 1995 to now), the *Transition Region and Coronal Explorer* [24] (*TRACE*; 1998–2010), the *Reuven Ramaty High Energy Solar Spectroscopic Imager* [25] (*RHESSI*; 2002–2018), the *Hinode* [26] (2006 to now), the *Solar Terrestrial Relations Observatory* [27] (*STEREO*; 2006 to now), the *Solar Dynamics Observatory* [28] (*SDO*; 2010 to now) and the *Interface Region Imaging Spectrograph* [29] (*IRIS*; 2013 to now). Also, more and more ground-based large-aperture solar telescopes have been put into routine observation; for example, the Swedish Solar Telescope [30] (SST; 1 m), the Goode Solar Telescope [31] (GST; 1.6 m) and the New Vacuum Solar Telescope [32] (NVST; 1 m). So far, the temporal (spatial size) resolution has been largely improved from tens of minutes (several arcseconds) to a few seconds (sub-arcseconds), and we now observe the Sun from multiple view angles with imaging and spectroscopic instruments covering a broad waveband from radio to hard X-ray (HXR). The current high spatio-temporal resolution imaging and spectroscopic and stereoscopic observations continuously increase our knowledge about solar jets. In addition, because of the tremendous improvement in computing power and calculation techniques, numerical modelling of solar jets has also made great progress in recent years. Nowadays, the study of solar jets has become a main research field in solar physics.

Over the past three decades, significant advances achieved in observational, theoretical and numerical analyses have contributed to shaping our evolving understanding of the different aspects of solar jets, such as their triggering and driving mechanisms fine structures as well as their relationship with other solar activities [33]. Now, we recognize that the basic energy release mechanism in solar jets is magnetic reconnection, which converts magnetic free energy into kinetic energy, thermal energy and radiant energy of particles [34,35]. Recent high spatio-temporal resolution observations showed that solar jets are probably miniature versions of large-scale eruptions such as filament eruptions and coronal mass ejections (CMEs) [36,37]. In addition, recent ultra-high-resolution observations further indicated that the eruption of small spicules is possibly the same as solar jets, since they were shown to be associated with magnetic flux cancellations and possibly mini-filament eruptions in several observational studies [9,13,38,39]. Therefore, these results may suggest a scale invariance of solar eruptions, and our understanding of solar jets can probably be applied to interpret the complicated and energetic large-scale solar eruptions and currently unresolved small spicules. Solar jets are also important for space weather forecasting, because they often eject large-scale mass and energetic particles into the interplanetary space.

Despite the great progress achieved in the past, the detailed physics behind solar jets is still not completely understood. For example, questions about their triggering and driving mechanisms,

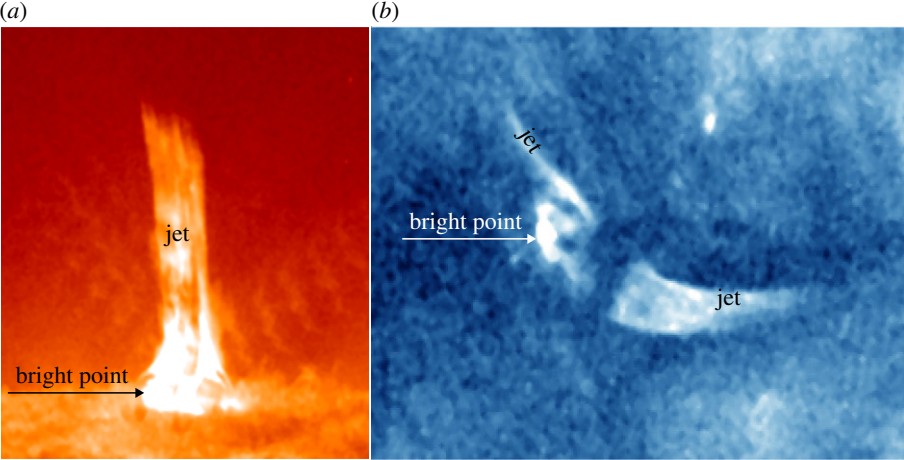

**Figure 1.** *SDO*/Atmospheric Imaging Assembly 304 Å (*a*) and 335 Å (*b*) images show an anemone jet on 30 January 2015 and a two-sided-loop jet on 2 June 2013 [44], respectively. The arrows indicate the bright points in the eruption source regions. (Online version in colour.)

evolution behaviours, fine structures and particle acceleration: How do small-scale solar jets evolve into large-scale CMEs? How do they contribute to coronal heating and solar wind acceleration? In addition, are different jetting phenomena from the photosphere to the low corona interconnected, or are they driven by the same mechanism but different emission components as the same basic physical process? Could the current models of solar jets be applied to explain small spicules and large filament/CME eruptions? The main aims of this review are to summarize our current knowledge of solar jets and to attempt to discuss how close we are to the answers to the above questions. Readers who are interested in the relevant research fields can refer to several previous reviews [16,33,40–42].

## 2. Observational feature

### (a) Morphology and classification

Solar jets are generally described as collimated, beam-like ejecting plasma flows along straight or slightly oblique magnetic field lines. Because of the huge improvement in observing capabilities, solar jets can be imaged in a wide temperature range from the photosphere to the outer corona. According to different classification methods, solar jets were classified into different types in history. Firstly, solar jets were divided into photospheric jets, chromospheric jets (or surges), transition region jets, coronal jets and white-light jets, based on the temperature of the atmosphere in which they occur. Secondly, they were classified as coronal hole jets, active region jets and quiet-Sun region jets, based on regions where they occur. Thirdly, they were classified as Hα jets, extreme ultraviolet (EUV) jets and X-ray jets, depending on different observing wavebands. Nevertheless, since solar jets are often observed simultaneously at different wavebands covering a wide temperature range, and they can occur in all types of solar regions, it seems that the above classification methods are not very reasonable if one considers the physical properties and morphologies.

Based on morphology, Shibata *et al.* [43] classified coronal jets into straight anemone jets and two-sided-loop jets (figure 1). An anemone jet exhibits an inverted-Y shape consisting of a straight plasma beam and a bright dome-like base. By contrast, a two-sided-loop jet appears as a pair of plasma beams ejecting in opposite directions from the eruption source region [44–49]. Recently, high-resolution observations combined with extrapolated three-dimensional (3D) coronal magnetic fields revealed the fan–spine topology magnetic system of straight anemone jets [50], which consists of a coronal nullpoint, a dome-like fan that represents the

closed separatrix surface, and inner and outer spines belonging to different connectivity domains [51–60]. A fan–spine topology often arises when a parasitic magnetic polarity emerges (or carries) into a pre-existing magnetic field region with opposite polarity, and a jet occurring within it can lead to three flare ribbons as a result of the low-altitude impact of the particle beams accelerated through the nullpoint magnetic reconnection; namely, an inner bright patch surrounded by a circular ribbon relevant to the inner spine and the dome-like fan structure, and a remote elongated bright ribbon associated with the outer spine. In principle, the fan–spine topology represents the 3D magnetic structure of all straight anemone jets. For straight jets in a coronal hole, their outer spines are very long and can be regarded as open fields in the outer corona; therefore, the remote footpoints (brightenings) of the outer spines cannot be identified, and these jets can be considered as eruptive jets. In comparison, for straight jets in or around active regions, one can often identify their entire fan–spine structures owing to their shorter outer spines. For these jets, they can be considered as confined jets, because their ejecting material is typically observed to be confined within the fan–spine system. Straight anemone jets could be further divided into inverted-Y and λ types, in which an inverted-Y (λ)-type jet was commonly interpreted as a small-scale magnetic bipole reconnecting with the ambient open coronal magnetic fields around the bipole top (footpoint). Hence, the different shapes could possibly be used to distinguish the reconnection sites in solar jets [61].

Moore *et al.* [62] classified straight anemone jets into standard jets and blowout jets based on their different physical properties. According to their definition, blowout jets exhibited several different distinguishing characteristics relative to standard jets that are the same as typical anemone jets, including (i) an additional bright point inside the base arch besides the outside one, (ii) a blowout eruption of the base arch that often hosts a twisting mini-filament, and (iii) an extra jet-spire strand rooted close to the outside bright point (figure 2). At the beginning, Moore *et al.* [62] found that about two (one)-thirds of anemone jets belong to standard (blowout)-type jets; however, this result was then updated to be about 50% each when the statistical samples were expanded [64]. Sterling *et al.* [36] studied 20 polar jets and proposed that all jets are generated in the same way as blowout jets; they argued that the successful (failed) eruptions of the mini-filaments inside the base arches can account for the observational characteristics of blowout (standard) jets. In a subsequent paper, Moore *et al.* [65] further examined 15 of the 20 jets studied by Sterling *et al.* [36] to determine the onset of the magnetic explosion in polar coronal jets; they found that a large majority of polar jets work in the same way as large-scale magnetic breakout eruptions in association with energetic flares and CMEs, in which the external breakout reconnection proceeds and is involved in the triggering of the eruption. Taken together with the results of Panesar *et al.* [66–68] they also claim that flux cancellation is the main process whereby the energy is stored prior to eruption in all jets and CMEs, and may also be involved in the triggering process. So far, the finding of blowout jets has been confirmed by many observational studies, and now we recognize that the vast majority of solar jets are caused by magnetic flux cancellation rather than flux emergence [63,69–84]. Recently, high-resolution observational studies have shown that two-sided-loop jets are also associated with flux cancellations and include the eruption of mini-filaments inside the base arches [44,46,49,85,86], and two-sided-loop jets occurring in filament channels may be important for causing the eruption of large filaments [87].

## (b) Precursor

It have been demonstrated that solar jets are launched from various pre-eruption structures, including satellite sunspots (or small opposite-polarity magnetic elements), mini-filaments, coronal bright points and mini-sigmoids. Observationally, these structures can be regarded as the progenitor of solar jets, and studying them can contribute to the prediction of the occurrence and evolution characteristics of solar jets.

In the photosphere, satellite sunspots can be considered as a conspicuous progenitor for surges or many active region coronal jets. Rust [20] reported that surges are liable to occur

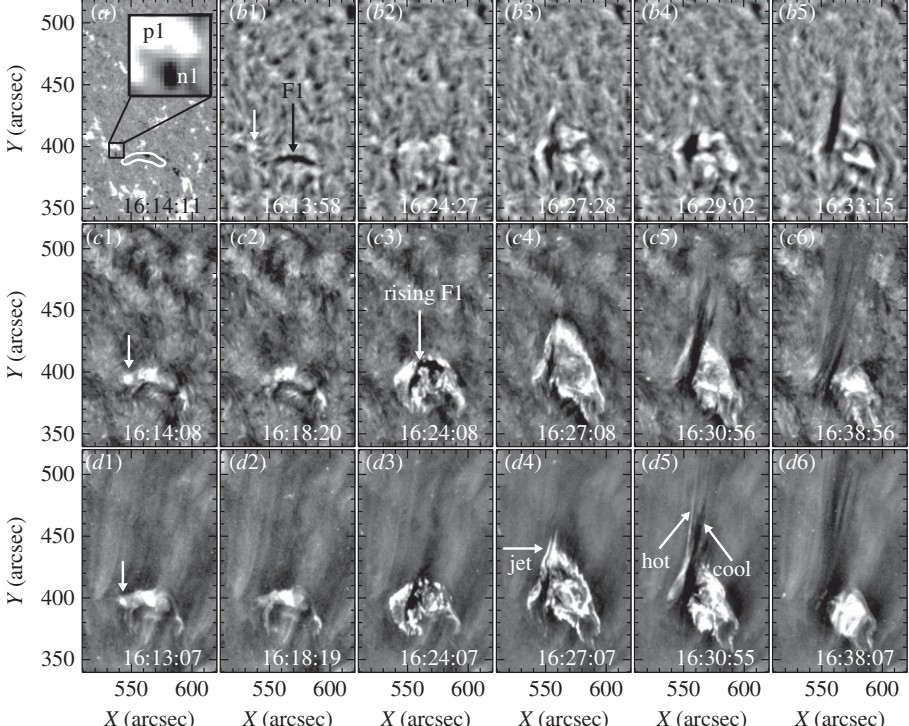

**Figure 2.** An example of blowout jets [63]. (*a*) is a *SDO*/Helioseismic and Magnetic Imager magnetogram, in which the inset shows a small bipole near the left end of the mini-filament (white contour). (*b*1)–(*b*5) are Hα images from the Big Bear Solar Observatory. (*c*1)–(*c*6) and (*d*1)–(*d*6) are *SDO*/Atmospheric Imaging Assembly 304 Å and 193 Å images, respectively. The white arrows in (*b*1), (*c*1) and (*d*1) point to a bright patch at the location of the small bipole prior to the jet. The black arrow in (*b*1) and the one in (*c*3) indicate the pre-eruption and rising phases of the mini-filament, respectively. The arrow in (*d*4) indicates the first appearance of the hot jet, and the two arrows in (*d*5) point to the jet's hot and cool components, respectively.

at nullpoints above satellite sunspots. Roy [21] confirmed this finding and further proposed that significant magnetic flux change over a short time interval is also important for producing surges. Subsequent studies based on high-resolution observations revealed that evolving satellite sunspots are liable to launch recurrent jets through continuous collisions with the main sunspots [55,88–94]. In quiet-Sun and coronal hole regions, small opposite-polarity magnetic elements can be recognized as the most conspicuous photospheric progenitor for many lower-energy, small-scale coronal jets. Generally, observations suggest that the onset of these jets is often closely associated with magnetic flux cancellations caused by the converging and/or shearing motions of the magnetic elements' opposite polarities [63,65,66,68].

Mini-filaments in the chromosphere and the corona can be recognized as an important progenitor for producing coronal jets. Mini-filaments were found to be eruptive in nature, and their eruption characteristics are similar to those evidenced in large-scale filament eruptions [95]. Several earlier observations showed that mini-filament eruptions are closely associated with coronal jets; however, the authors did not clarify the physical relationship between them [89,96]. Shen *et al.* [63,69] studied two blowout jets and found that the erupting mini-filaments directly form the cool component of coronal jets (figure 2). Recently, more and more observational studies have confirmed that blowout jets are driven by mini-filament eruptions or filament channels [73–84,97,98]; some authors even proposed that all coronal jets originate from mini-filament eruptions, and their generation resembles the eruption of large-scale, energetic filament/CME eruptions [36,65,99].

Solar jets are frequently observed to be ejected from coronal bright points and micro-sigmoids. Coronal bright points represent a set of small-scale low-corona loops with enhanced emission in

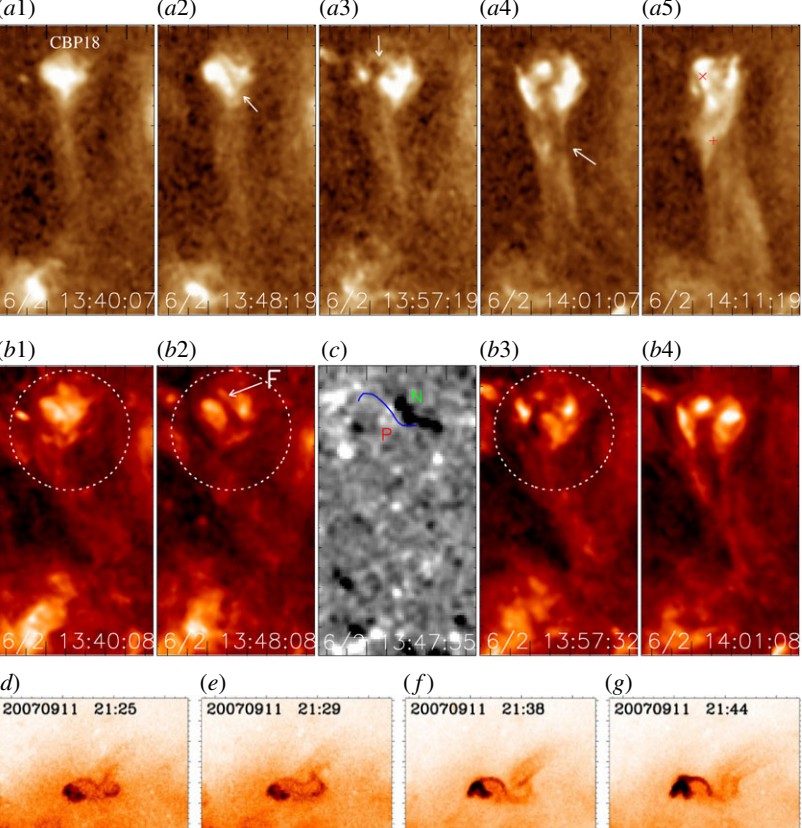

**Figure 3.** Examples of solar jets originated from EUV bright points [74] (top and middle rows) and mini-sigmoid structures [110] (bottom row). (a1–a5), (b1–b4) and (c) are *SDO*/Atmospheric Imaging Assembly 193 Å, 304 Å and magnetogram images, respectively. A mini-filament is indicated by the arrows in (a2) and (b2), and the jet spire is indicated by the arrow in (a4). The circles mark the bright point region. The blue curve in (c) indicates the position of the mini-filament on the magnetogram. (d)–(g) are *Hinode* soft X-ray images from the *Hinode*. (Online version in colour.)

the EUV and X-ray spectra [100,101], and plasma ejections were found to originate from them [6,102,103]. Recent high-resolution observations showed that coronal bright points are ubiquitous in the corona [104], and they were found to be liable to produce solar jets [105–108]. Statistical studies indicated that a majority of coronal bright points produce at least one eruption during their lifetime ($\approx$21 h) [109]. Hong *et al.* [74] found that about one-quarter to one-third of coronal bright points produce one or more filament-driven blowout jets during their lifetimes (see the top and middle panels of figure 3).

A coronal sigmoid consists of many differently oriented loops that all together form two opposite J-shaped bundles or an overall S-shaped structure [111], which is more likely to be eruptive and is the main progenitor of solar eruptions [112]. Micro-sigmoids have a typical size of about one-fifth of the large-scale ones; they can be formed through injecting twists into simple coronal bright points [113], or via a tether-cutting reconnection mechanism [108]. Using the *Hinode* X-ray observations, Raouafi *et al.* [110] identified that some coronal jets evolve from mini-sigmoids (see the bottom panels of figure 3). Liu *et al.* [114] reported a special case in which a pair of twin blowout jets were successively generated from a sigmoid structure; the authors proposed that the two jets were produced by the reconnection between the ambient open fields and the two opposite J-shaped twisted sigmoidal magnetic fluxes, respectively. These observations indicate that coronal mini-sigmoids can be recognized as a progenitor of coronal jets [110].

## (c) Fine structure

### (i) Cool and hot components

Sometimes, cool and hot plasma flows can be identified simultaneously in a single jet. The co-existing cool and hot components can be observed in EUV images, or separately appear in Hα images as a surge and in EUV or soft X-ray images as a coronal jet [63,69,88,115–117]. Some earlier studies indicated that the cool and hot components of solar jets are correlated in time and space, and with similar kinematic behaviours [118–120]. The co-spatial relationship was confirmed by some recent high-spatio-temporal-resolution observations [121], although the two components were not exactly co-spatial over the entire length [115], or the hot component had a higher speed than the cool one [122]. Nevertheless, the two components of solar jets were also found to be adjacent to each other in some events [63,69,88,96,117,123,124]. Particularly, Chae *et al.* [123] reported several jets whose hot EUV components were identified with the cool Hα bright jet-like features. Jiang *et al.* [117] studied three jets whose cool and hot components showed different evolutions not only in space but also in time, in which the cool Hα component had a smaller size than the larger, hot one, and the former moved along the edges of the latter.

The cool component in a jet often appears a few minutes after the corresponding hot one [63,69,115,117,125–127]. Specially, in the outer corona at a height of 1.71 $R_\odot$, Dobrzycka *et al.* [128] observed five polar jets in which the cool components arrived about 25 min after the hot ones. There is one case in which the cool component (surge) appeared about 2 h before the corresponding hot X-ray jet [90]. This abnormal result was probably caused by the unsteady cadence and low spatial resolution of the *Yohkoh* X-ray data. Since solar jets can occur repeatedly from the same source region, and their lifetimes are usually of 40 min or less according to high-resolution observations, the 2 h time interval seems too long for a single jet. Hence, the cool surge and the hot X-ray jet in Zhang *et al.* [90] were very possibly two different jets that originated from the same source region, rather than simultaneous cool and hot components in a single jet.

The different spatial relationships can possibly be reconciled by considering the projection effect. In principle, observational results suggested that the two components are along different magnetic field lines and dynamically connected. Therefore, their spatial relationship should not be co-spatial but adjacent to each other. The co-spatial case can be expected when the two components overlap each other along the line of sight. For the delayed appearance of the cool component, one can understand it based on the formation mechanism of solar jets. Previously, the delayed appearance of the cool component was explained as the cooling of the earlier, hotter one [115,117,125], the emerging chromospheric or transition region cool plasma accelerated the magnetic tension force of the newly formed magnetic reconnection field lines [123,127,129] or the different Alfvén speeds in the cool (high-density, low Alfvén speed) and hot (low-density, high Alfvén speed) plasma flows [130]. Based on high-spatio-temporal-resolution observations (figure 2*d*5), several works have provided evidence that the cool component of jets is directly formed by the erupting mini-filaments confined within the jet base [36,38,39,44,62–64,69,74,109, 131]. According to the interpretation of Shen *et al.* [44,63,69], a mini-filament is confined by a small arch surrounded by open field lines. For some reason, the arch starts to reconnect with the ambient open field lines (external reconnection), which produces the hot jet component like the generation of a standard jet. The external reconnection not only produces the hot jet but also removes the confining field lines of the mini-filament, which further results in the instability and eruption of the filament owing to the internal reconnection between the two legs of the confining field lines. Therefore, the appearance of the cool component is naturally after the hot one, and their spatial relationship is adjacent to each other. So far, more and more observations have indicated the appearance of a cool component in solar jets, especially in blowout jets that often involve the eruption of mini-filaments [36,38,39,62,64,74,109,131].

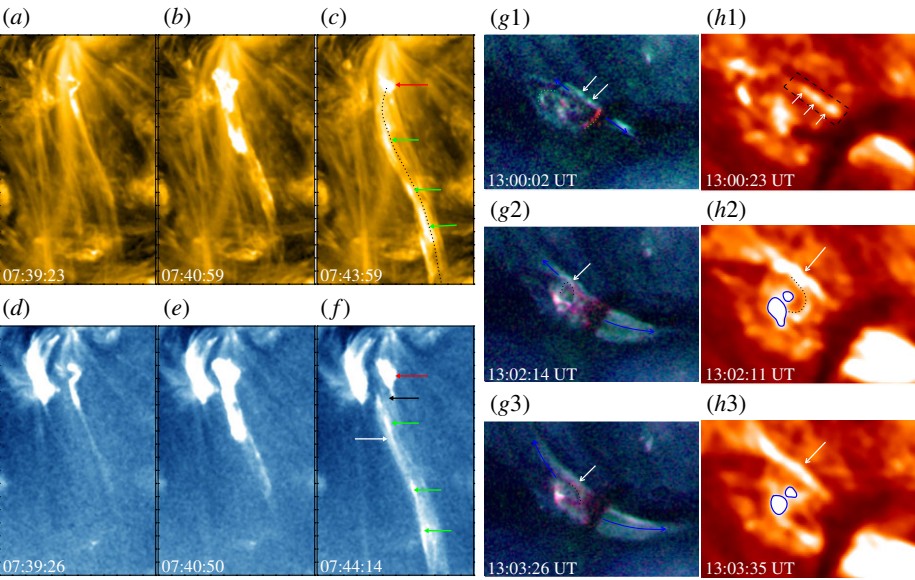

**Figure 4.** Examples of plasmoids in anemone [69] (*a–f*) and two-sided-loop jets [44] (*g*1–3,*h*1–3). The anemone jet is displayed with the *SDO* 171 Å (*a–c*) and 335 Å (*d–f*) images, in which the green arrows indicate the plasmoids along the jet spire. The two-sided-loop jet is shown with the *SDO* composite high-temperature images made from the Atmospheric Imaging Assembly 94 Å (red), 335 Å (green) and 193 Å (blue) channels (*g*1–3) and the 304 Å (*h*1–3) images. The arrows in the top row point to the plasmoids in the current sheet, while those in the middle and bottom rows indicate the current sheet. (Online version in colour.)

### (ii) Plasmoid

Supported by various observational indications [40], it has been widely accepted that solar jets are produced by magnetic reconnection [33]. A typical indication supporting the reconnection scenario is the formation and ejection of outward plasmoids (also called magnetic islands, or plasma blobs) due to the tearing-mode instability of the current sheets. Singh *et al.* [132] observed multiple bright plasmoids in several jets imaged by the Ca II *H* filtergram of the *Hinode*/Solar Optical Telescope (SOT), whose typical lifetime, size and intensity enhancement relative to the background are about 20–60 seconds, 0.3–1.5 Mm and 30%, respectively. Plasmoids have a multi-thermal (1.4–3.4 MK) structure with a number density in the range of $1.7$–$2.89 \times 10^{10}$ cm$^{-3}$ [69,133,134]. Subarcsecond plasmoids were detected by the *IRIS*, whose average size is about 0.57 Mm, and their ejecting speed ranges from 10 to 220 km s$^{-1}$ [135]. Besides the observation of plasmoids in straight anemone jets [81,136–139], they were also observed in two-sided-loop jets [44] (figure 4). Analysis results indicated that plasmoids observed in different types of jets were similar, and they were thought to be created by magnetic reconnection as a result of the tearing instability of the current sheets.

Numerical experiments also revealed the appearance of plasmoids in solar jets. Yokoyama & Shibata [129,140] performed the two-dimensional (2D) simulations of solar jets, in which they evidenced the creation, coalescence and ejection of plasmoids in the current sheets. In another simulation, high-temperature and high-density plasmoids were generated repeatedly at the same location and were ejected upwards and downwards simultaneously [141]. The authors claimed that the merged upward-moving plasmoids correspond to the anemone jets, as in observations. Ni and co-workers [142,143] tested the low- and high-plasma $\beta$ cases to study the formation of plasmoids in solar jets; they found that plasmoids with similar characteristic parameters to those observed are easily created in the low $\beta$ case, while the high $\beta$ case created vortex-like structures as a result of Kelvin–Helmholtz (KH) instability. According to this study, the observed discrete high-density features in solar jets could be either plasmoids or vortex structures at different

wavebands. In 3D simulations, plasmoids were evidenced as twisted flux ropes resembling the shape of solenoids, and they are most likely formed as a result of the resistive tearing-mode instabilities in the current sheets located between closed and open fields [144,145]. Wyper *et al.* [146] pointed out that the tearing process should occur at the separatrix surface between the closed and open flux systems, and the repeated formation and ejection of flux ropes can naturally explain the intermittent outflows, bright blobs and filamentary structures observed in some jets.

### (iii) Kelvin–Helmholtz instability

KH instability is a basic physical process that occurs when there is velocity shear in a single continuous fluid, or when there is a velocity difference across the interface between two fluids [147]. Recent observations have revealed the occurrence of KH instability in the solar atmosphere, such as at the interface between an erupting region and the surrounding corona [148], at the outer edge of CMEs [149], in prominences [150], in coronal streamers [151] and in solar jets [152,153].

Vortex structures caused by KH instability can be regarded as a basic fine structure of solar jets, which were frequently observed within or at the outer edges of solar jets [154,155]. Using *IRIS* observations, Li *et al.* [152] reported the developing process of KH instability in a blowout jet due to the strong velocity shear of two plasma flows along the jet spire, in which the developing process was about 80 s and the distortion scale was less than 1.6 Mm. Using Hα observations taken by the NVST, Yuan *et al.* [153] studied the KH instability at the outer edge of a small solar jet, in which the KH instability was thought to be caused by the shearing motion between cool chromospheric and hot coronal plasma flows. During the mature stage, plasma heating was evidenced around the region of the vortex structures, supporting the scenario that KH instability can effectively transfer plasma kinetic energy into thermal energy and heat the coronal plasma. Since velocity shear can occur at a variety of length scales and different regions in the solar atmosphere, this finding led the authors to conjecture that KH instability could be an effective way to supply energy to heat the corona plasma.

Theoretical and numerical works were performed to study the KH instability in rotating solar jets. Zaqarashvili *et al.* [156] found that rotating jets are unstable to KH instability when the kinetic energy of rotation is more than the magnetic energy of the twist; the growth time of KH instability is several seconds for miniature jet-like events and a few minutes or less for large jets. The authors argued that rotating jets may provide energy for chromospheric and coronal heating, since KH vortices can lead to enhanced turbulence development and heating of the surrounding plasma.

## (d) Dynamic characteristics

### (i) General properties

Based on *Yohkoh* soft X-ray observations, Shimojo *et al.* [157] described several typical properties of X-ray jets, including that (i) most of the jets are associated with micro-flares whose brightest parts show a gap between the exact footpoints of the jets; (ii) the lengths (widths) are in the range of a few ten thousands of kilometres to $4 \times 10^5$ ($5 \times 10^3$–$10^5$) kilometres; (iii) the apparent velocities are of 10 to 1000 km s$^{-1}$ with a mean value of about 200 km s$^{-1}$; (iv) the distribution of the lifetimes is a power law with an index of about 1.2; (v) most active region jets are observed to the west of the active regions; (vi) 76% of jets show constant or converging spires whose widths get narrower from the photosphere to the corona, and their intensity distribution often shows an exponential decrease with distance from the footpoints. In a subsequent paper [4], they further concluded that (i) the temperatures of the jets are 3–8 MK with an average value of 5.6 MK, similar to those of the associated flares, and this shows a correlation with the sizes of the associated flares; (ii) the density is in the range of 0.7−4.0 × 10$^9$ cm$^{-3}$ with an average value of $1.7 \times 10^9$ cm$^{-3}$; (iii) the thermal energies of the jets are $10^{27}$−$10^{29}$ ergs, far less than those of the associated flares; and (iv) the apparent velocity of the jets is usually

slower than the speed of sound. The physical parameters were further studied based on a large sample of 7197 coronal hole X-ray jets observed by the *Hinode* [158]. The authors found that the peaked distributions with maxima of the outward velocities, the lengths, widths and lifetimes of the jets were $160\,\mathrm{km\,s^{-1}}$, $5 \times 10^4\,\mathrm{km}$, $8 \times 10^3\,\mathrm{km}$ and 10 min, respectively. In addition, the velocities of the transverse motions perpendicular to the jet axis ranged from 0 to $35\,\mathrm{km\,s^{-1}}$.

Using Ca II *H* broadband filter observations taken by the *Hinode*/SOT, Nishizuka *et al.* [159] made a statistical study of chromospheric anemone jets [1]. Different from spicules [15,16] and dynamic fibrils [160], chromospheric anemone jets are usually observed in active regions and show bright cusp-like or inverted Y-shaped structures, and are smaller and occur much more frequently than surges. The authors found that the shape of chromospheric anemone jets is similar to that of X-ray jets, suggesting a common formation mechanism. The typical parameters of lengths, widths, lifetimes and velocities of chromospheric anemone jets are in the ranges of about 1–4 Mm, 100–400 km, 100–500 s and $5$–$20\,\mathrm{km\,s^{-1}}$, respectively. In addition, the velocities are found to be comparable to the local Alfvén speed in the lower solar chromosphere.

Nisticò *et al.* [61] performed a statistical study of energetic polar EUV jets using *STEREO* observations; they found that the appearance of EUV jets is always correlated with small-scale chromospheric bright points. The typical lifetimes of the studied EUV jets are 20 (30) min at 171 (304) Å, while those of the white-light jets observed in coronagraphs peak at around 70–80 min. It was found that the speeds were 400 and $270\,\mathrm{km\,s^{-1}}$ for the hot 171 Å and cool 304 Å components, respectively. The speeds measured from 171 Å observations are comparable to those derived from coronagraph observations ($390\,\mathrm{km\,s^{-1}}$). Mulay *et al.* [161] studied active region EUV jets observed by the *SDO*; their results indicated that the lifetimes and velocities are in the ranges of 5–39 min and $87$–$532\,\mathrm{km\,s^{-1}}$, respectively, and the corresponding average values are 18 min and $271\,\mathrm{km\,s^{-1}}$, respectively. Typically, all the studied jets were co-temporally associated with Hα jets and non-thermal type III radio bursts, and 50% (30%) of the events in their samples originated in the regions of flux cancellation (emergence). Other similar statistical studies based on *STEREO* and *SDO* observations can also be found in the literature [162,163]. In statistical studies of white-light jets based on coronagraphs onboard the *SOHO*, the speeds of jets at the solar minimum of activity are in the range of $400$–$1100\,\mathrm{km\,s^{-1}}$ for the leading edge and $250\,\mathrm{km\,s^{-1}}$ for the bulk of their material, while the typical speeds at the maximum of activity are around $600\,\mathrm{km\,s^{-1}}$ [164,165]. In addition, Kiss *et al.* [166] carried out a statistical study of 301 macrospicular jets using the Atmospheric Imaging Assembly (AIA) 304 Å observations from June 2010 to December 2015. The authors found a strong asymmetry in the spatial distribution in terms of solar North/South hemispheres, and the average lifetime, width, length and velocity of the studied macrospicular jets were $16.75 \pm 4.5$ min, $6.1 \pm 4$ Mm, $28.05 \pm 7.67$ Mm and $73.14 \pm 25.92\,\mathrm{km\,s^{-1}}$, respectively.

### (ii) Rotating motion

Rotating motion is a typical dynamic characteristic of solar jets. Earlier observations demonstrated the appearance of rotating motion in prominences like a tornado [167–169], and this kind of motion is frequently seen in erupting filaments [170–176]. Xu *et al.* [177] detected the rotating motion in a surge using the Hα spectral observations. With the improvements in the quality of imaging and spectral observations, rotating motion was widely observed in Hα surges [90,178,179], EUV jets [2,88,115,117,180,181] and macrospicules [182,183]. Some observations suggested that the eruptions of twisted filaments sometimes show as rotating jets [84,174,184], and this can be explained as a result of the reconnection between twisted filaments and their surrounding open fields [185]. In such a process, the magnetic twists stored in the closed filaments or loops are released into open magnetic fields, and plasma is driven out in the relaxation process [2,88,179].

Stereoscopic observations taken by *STEREO* imaged the fine helical structure of the rotating jets, which exhibited different morphologies when they were observed from different viewing angles [180]. A statistical study based on *STEREO* observations indicated that at least half of EUV

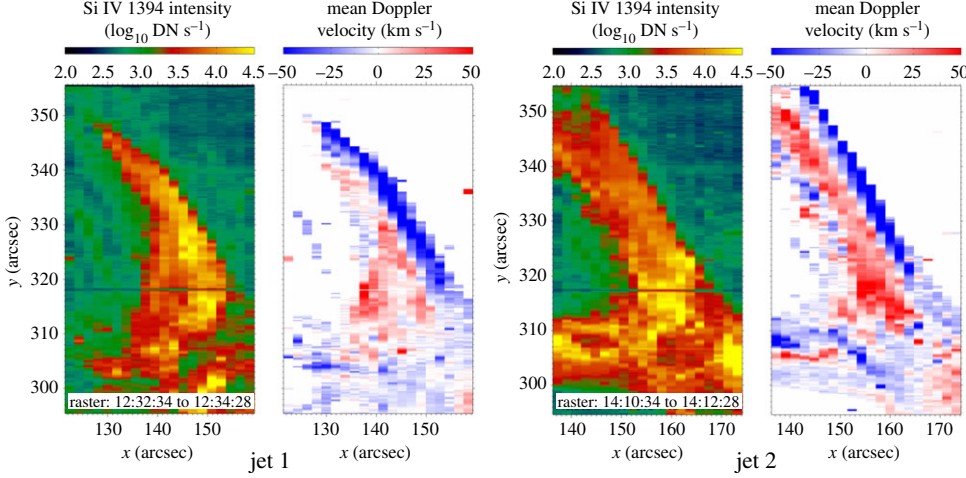

**Figure 5.** Spectroscopic observations of two rotating jets [194]. For each jet, the total intensity and mean Doppler velocity are computed from the *IRIS* Si IV 1394 line. In the Doppler velocity maps, blue- and redshift signals are prominent around the northern and southern edges of the jets. (Online version in colour.)

jets exhibited a helical magnetic field structure [61]. Using the *SDO* data, Shen *et al.* [2] studied a rotating polar coronal hole jet which exhibited distinct bright helical structures around the jet axis; the authors proposed that the rotating jet was driven by the release of the magnetic twist stored in the pre-existing arch into the ambient open field through magnetic reconnection. Measurement results indicated that the period of rotation and the twists transferred into the open fields were about 564 s and 1.17–2.55 turns, respectively. Specifically, the statistical studies carried out by Moore *et al.* [64,186] found the existence of obvious axial rotation in both standard and blowout jets; their results showed that the number of turns of axial rotation ranged from 0.25 to 2.5. Other case studies showed that the number of turns of axial rotation ranged from about 0.34 to 4.7 [187–190], and the periods were measured to be 1–20 min [174,191,192].

The rotating motion of solar jets manifests as simultaneous blue- and redshift on either side of the jet body in Doppler velocity maps [88,107,181–183,193]. Pike & Mason [182] reported the appearance of simultaneous blue- and red-shifted emissions on either side of macrospicules, which was interpreted as the presence of rotation plasma flows. Kamio *et al.* [183] reported similar Doppler velocity patterns and measured the blue- and redshift as about $-120$ and $50 \, \mathrm{km \, s^{-1}}$, respectively. Cheung *et al.* [194] studied four homologous helical jets at transition region temperatures, which showed evidence of oppositely directed flows with components reaching Doppler velocities of $\pm 100 \, \mathrm{km \, s^{-1}}$, and the magnetic twists needed for the helical jets were found to be supplied by emerging current-carrying magnetic fields (figure 5). Lu *et al.* [195] studied a recurrent jet event using spectroscopic and stereoscopic observations, in which the Doppler velocities were about $\pm 90 \, \mathrm{km \, s^{-1}}$, which is consistent with the value derived from stereoscopic imaging observations. A statistical study performed by Kayshap *et al.* [196] indicated that rotational motion is omnipresent in network jets, which can be detected as blueshift on one edge and redshift on the other at a mean rotational velocity of about $49.56 \, \mathrm{km \, s^{-1}}$.

### (iii) Transverse motion

Lateral expansion is a typical characteristic of solar jets, which manifests as the whip-like upward motion of the newly formed field lines [88,129]. The typical expansion speeds were found to be tens of kilometres per second [105,158,177]. Savcheva *et al.* [158] found that most X-ray jets exhibited lateral motion, which showed some acceleration and deceleration before and after a

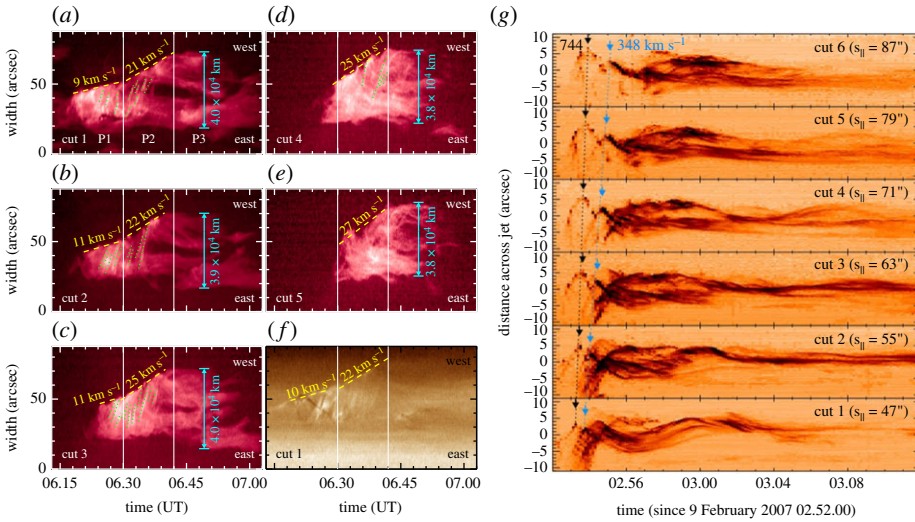

**Figure 6.** Examples of lateral expansion [2] (*a–f*) and transverse oscillation [199] (*g*) of solar jets, shown with time–distance plots perpendicular to the main axis of the jets at different heights. (*a–e*) and (*f*) are from the *SDO* 304 Å and 193 Å observations, respectively. The expanding speeds and final width of the jet are indicated. The panels in (*g*) are from the *Hinode* Ca II H images. The black and blue arrows point to the crests of the first two inverted-V-shaped tracks. The dotted lines are linear fits to the propagation of the two crests, and the derived phase speeds are marked. (Online version in colour.)

period of constant lateral motion. Shimojo *et al.* [197] reported the successive slow ($10 \, \text{km s}^{-1}$) and fast ($20 \, \text{km s}^{-1}$) lateral expansions in an X-ray jet, in which the slow expanding stage was explained as the loop escaping from the anti-parallel magnetic field, while the fast stage corresponded to the whip-like motion of the reconnected field lines. By contrast, Chandrashekhar *et al.* [198] proposed that the progressive reconnection occurring in magnetic structures along the neutral line could account for the slow lateral expansion motion of solar jets. Similar slow ($16 \, \text{km s}^{-1}$) and fast ($135 \, \text{km s}^{-1}$) expanding motions of loop systems were also observed in small chromospheric anemone jets [191], in which the transition from the slow to the fast expansion stage occurred at the start of the accompanying flares.

Shen *et al.* [2] reported an EUV jet in which the lateral expansion showed three distinct stages: slow ($10 \, \text{km s}^{-1}$), fast ($25 \, \text{km s}^{-1}$) and constant stages. Both the slow and fast expansion stages lasted for about 12 min, and the jet kept a constant width of about $4 \times 10^4 \, \text{km}$ during the constant stage (figure 6*a–f*). The fast transition from the slow to the fast expansion stage was explained as the sudden acceleration of the magnetic reconnection between the emerging arch and the ambient open field. In other words, the slow expansion stage corresponded to the emerging period of the arch, during which its reconnection with the ambient open field was slow, while the fast expansion stage manifested as the impulsive reconnection between the two magnetic systems. The constant stage indicated the full opening of the closed arch and the end of the twist transfer into the open fields, and its width corresponded to the distance between the footpoints of the open field line and the remote footpoint of the closed arch. In a statistical study [64], the authors found that all blowout jets showed obvious lateral expansions but none in standard jets; the lack of lateral expansion in standard jets was possibly due to the smaller lateral plasma pressure in the jets than the magnetic pressure of the surrounding open field lines.

Transverse oscillation is another distinct characteristic of solar jets (figure 6*g*). Cirtain *et al.* [7] proposed that the transverse oscillation manifests as the formation of Alfvén waves during the relaxation of the reconnected magnetic fields. Using a magneto-seismology technique, the transverse oscillations of solar jets were used to derive some important physical parameters. For example, considering the transverse oscillation as a kink mode oscillation [200], Morton

*et al.* [201] estimated the temperature of a dark thread in a jet to be 2–3 × 10⁴ K; therefore, the authors proposed that the dark thread was likely to have originated in the chromosphere. Using the measured wave parameters of a on-disc coronal hole jet and the magneto-seismological inversion technique, Chandrashekhar *et al.* [202] estimated the magnetic field strength along the jet spire to be about 1.2 Gauss. The speed and period of the transverse oscillations were measured to be, respectively, about 100–800 km s$^{-1}$ and 200–536 s [7,191,199,201–203], consistent with the theoretical prediction results of Alfvén waves in solar jets [7,88].

# 3. Relation to other phenomena

## (a) Plumes

Coronal plumes are thin ray-like structures that are pervasive within polar and equatorial coronal holes, as well as quiet-Sun regions [204–206]; they are rooted in chromospheric networks and can be identified over distances of several solar radii, and even in the interplanetary space. Solar jets show some common properties with plumes; for example, both are collimated magnetic structures resulting from magnetic reconnections between closed and open field lines [207,208].

Lites *et al.* [209] observed an EUV jet that was embedded in a polar plume and that caused notable density fluctuations within the plume structure. Ubiquitous episodic jets rooted in magnetized regions of the quiet corona were detected in plumes and interplume regions [205]. Raouafi *et al.* [210] studied 28 jets during the deep solar minimum and found that over 90% of the jets in their sample were associated with plumes, of which about 70% were followed by the formation of plumes with a time delay of minutes to hours, while the remaining jets occurred in pre-existing plumes and caused brightness enhancement of the latter. Therefore, the authors proposed that solar jets are precursors of plumes. In addition, short-lived, jet-like events and transient bright points were identified at different locations within the base of pre-existing long-lived plumes, which was thought to be important for the maintenance and change of the plume brightness. Raouafi & Stenborg [211] further found a large number of short-lived small jets and transient bright points caused by quasi-random cancellations between the minority magnetic polarity and the ambient dominant open magnetic fields, confirming their previous finding that plumes are dependent on the occurrence of transients resulting from low-rate magnetic reconnection. However, solar jets may not be a necessary step for the formation and maintenance of plumes, because not all jets are accompanied by the formation of plumes, and the birth of plumes follows the occurrence of jets. Therefore, the relationship between jets and plumes needs further in-depth investigations.

## (b) Filaments

Solar jets are tightly associated with filaments. On the one hand, as has been discussed in §§2b and 2ci, many solar jets result from mini-filament eruptions, and the erupting filament material forms their cool component. On the other hand, solar jets can cause the oscillation, formation and eruption of large-scale filaments [212–215].

Luna *et al.* [214] reported a case of large-amplitude longitudinal oscillation in a filament that was triggered by episodic jets along the filament axis; they proposed that the restoring force of the large-amplitude longitudinal filament oscillations was solar gravity, while the damping mechanism was the ongoing accumulation of mass onto the oscillating filament threads [216]. A similar event was reported by Awasthi *et al.* [217], in which the damping of the longitudinal filament oscillation was explained by the continued mass accretion supplied by the associated jets. Zhang *et al.* [215] reported the simultaneous transverse and longitudinal oscillations in a quiescent filament triggered by a coronal jet. Simultaneous transverse and longitudinal oscillations in filaments can also be excited by EUV waves [218]; it was found that the angle between the incoming waves and the filament axis is important to trigger the type of oscillations that appear [218,219]. It is thought that this can be used in the generation of simultaneous transverse

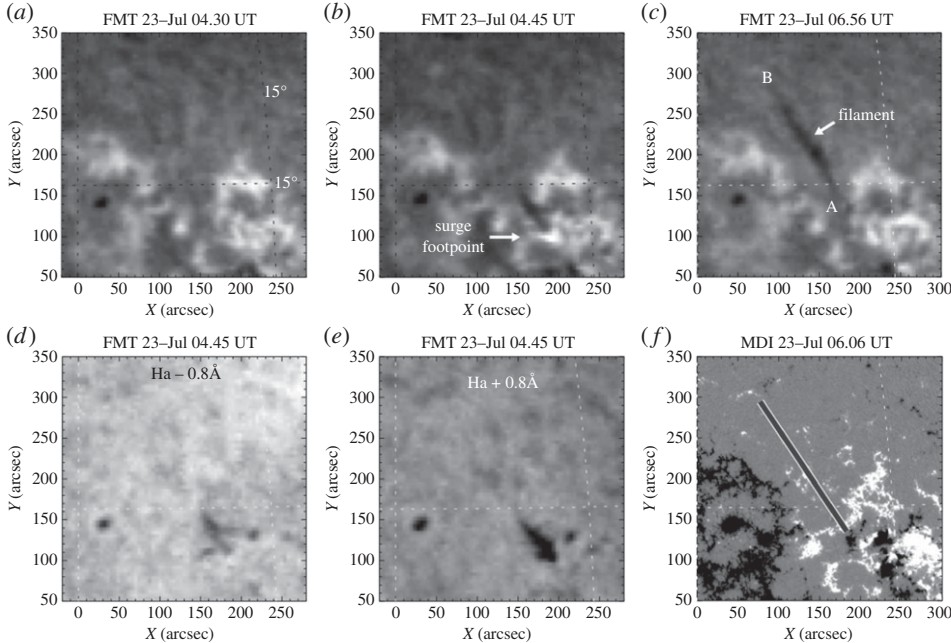

**Figure 7.** The formation of a filament owing to the injection of jet material [213]. (*a–c*) Hα-centre images from the Flare Monitor Telescope; (*d,e*) blue- and red-wing images at ±0.8 Å from the Hα centre. The arrows in (*b*) and (*c*) indicate the surge and the newly formed filament, respectively. (*f*) A *SOHO* magnetogram overlaid with the outline of the filament.

and longitudinal filament oscillations caused by jets. If a jet interacts with a filament along (perpendicular) the filament axis, large-amplitude longitudinal (transverse) oscillations can be expected; if the jet interacts with the filament with an acute angle with respect to the filament axis, simultaneous longitudinal and transverse oscillations can be launched.

Solar jets not only supply sufficient mass for filament formation but also cause the instability and eruption of large-scale filaments. Zirin [212] reported the formation of a short-lived filament caused by a surge through filling a semi-stable magnetic trap. Liu *et al.* [213] reported several similar events and found that the newly formed filaments exhibited distinct helical structures, whose lifetimes and average lengths were more than 20 h and 145 Mm, respectively (figure 7). The authors proposed two necessary conditions for new filament formation by jets; namely, an empty filament channel (or magnetic trap) and enough mass supplied by surges. Guo *et al.* [220] studied the formation and eruption of a large filament associated with a recurrent surge event; they confirmed that surge activities can efficiently supply enough mass for filament formation, and continuous mass with momentum supplied by surges can result in the instability and even the eruption of the newly formed filament. Other similar studies suggest that the material for filament formation could be supplied by both cool surges and hot coronal jets [221,222]. All the above studies showed that jet material was injected into filaments from one end of the filament channels. Recently, two-sided-loop jets in filament channels were found to be important in the maintenance of mass and the eruption of large-scale filaments. Shen *et al.* [44] reported that a two-sided-loop jet ejects material into an overlying large filament from below through magnetic reconnection, which provided an alternative way to understand how jet material injects into filament structures. In a recurrent two-sided-loop jet event in a filament channel, Tian *et al.* [87] found that the first jet caused the splitting of an overlying large filament into a double-decker filament, and then the following jets finally led to the full eruption of the filament. These studies showed the close relationship between solar jets and filaments, but detailed physical connection between them still needs further observational and theoretical investigations.

## (c) Magnetohydrodynamic waves

Solar jets are closely related to magnetohydrodynamic (MHD) waves. Observational studies indicated that solar jets can act as a driver to excite torsion Alfvén waves in themselves (see §2d(iii)), kink waves in remote coronal loops and filaments, and large-scale EUV waves. Statistical analysis indicated that the most probable mechanism for exciting kink oscillations in coronal loops is the deviation of loops from their equilibrium by nearby eruptions of plasma ejections [223,224]. Using the magneto-seismology technique, the measured oscillation parameters could be used to derive the magnetic field strength of the loops/filaments. For example, Sarkar *et al.* [225] reported a case of jet-driven transverse oscillation of a coronal loop whose magnetic field strength was estimated to be about 2.68–4.5 Gauss. Luna *et al.* [214] estimated the minimum magnetic field strength of an oscillating filament to be about 14 Gauss. Zhang *et al.* [215] derived the curvature radius of the long arcade supporting the filament to be about 244 Mm.

Although many previous studies have shown that large-scale EUV waves are driven by CMEs [226–230], recent observations suggest that they can also be launched by solar jets directly or indirectly. Shen *et al.* [231] reported that large-scale non-CME-associated EUV waves were excited by the sudden lateral expansion of transequatorial loops owing to the impingement of solar jets, in which the generation of the waves was caused by the sudden increase of gas and magnetic pressures around the expansion section of the loop. In a subsequent study, they further reported the generation of recurrent fast-mode EUV waves ahead of homologous jets along a large-scale transequatorial loop system (figure 8); they explained that the excitation mechanism of these waves resembles the generation mechanism of a piston shock in a tube [94]. Li *et al.* [232] also reported a nonlinear shock wave in a closed-loop system driven by a coronal jet at one of the footpoints of the loop; the authors proposed that such a kink in the wave can quickly heat the corona plasma through the rarefaction wave. Simultaneous EUV waves, quasi-periodic fast-propagating waves and kink waves were found to be launched during the interaction of a jet with a coronal loop [224]. In addition, the expansion of the strongly curved reconnected loops in solar jets can also launch large-scale EUV waves [233]. A typical characteristic of these non-CME-associated EUV waves is that their lifetimes (a few minutes) are much shorter than those driven by CMEs [231]. This is possibly because the transient solar jets cannot provide continuous driving to EUV waves like those driven by CMEs. EUV waves were shown to be important in triggering sympathetic solar activities, and it was observed that coronal hole jets could be launched by the passing of EUV waves [234]. This suggests that solar jets can also be produced by external disturbances except for internal magnetic activities such as flux cancellations and mini-filament eruptions.

## (d) Coronal mass ejection

CMEs represent large-scale plasma and magnetic fields being released from the Sun into the interplanetary space [235–237]. CMEs with an apparent angular width of $15°$ or less are typically associated with solar jets [238]. Observations suggested that solar jets cause not only narrow jet-like CMEs [164] (figure 9a), but also standard broad CMEs with a typical three-part structure [240]. Sometimes, paired narrow and broad CMEs can be simultaneously launched by a single blowout jet [63,239] (figure 9b).

Narrow jet-like CMEs are simply the outward extensions of solar jets in the outer corona [94,164]. Statistical studies showed that during a solar activity minimum the leading edges of the jet-like CMEs propagate at speeds of $400$–$1100\,\mathrm{km\,s^{-1}}$, while the bulk of their material travels at average speeds of about $250\,\mathrm{km\,s^{-1}}$ at heliocentric distances of 2.9–$3.7\,R_\odot$ [164]. By contrast, narrow jet-like CMEs during a solar activity maximum have a typical speed of about $600\,\mathrm{km\,s^{-1}}$, and they tend to be brighter and wider than those in a solar activity minimum [165]. The propagation of jet-like CMEs is not regulated by gravity, since some of them exhibit accelerations rather than decelerations above $3\,R_\odot$ [241]. In addition, the direction of CMEs originating from solar jets can be significantly changed through interactions with other magnetic

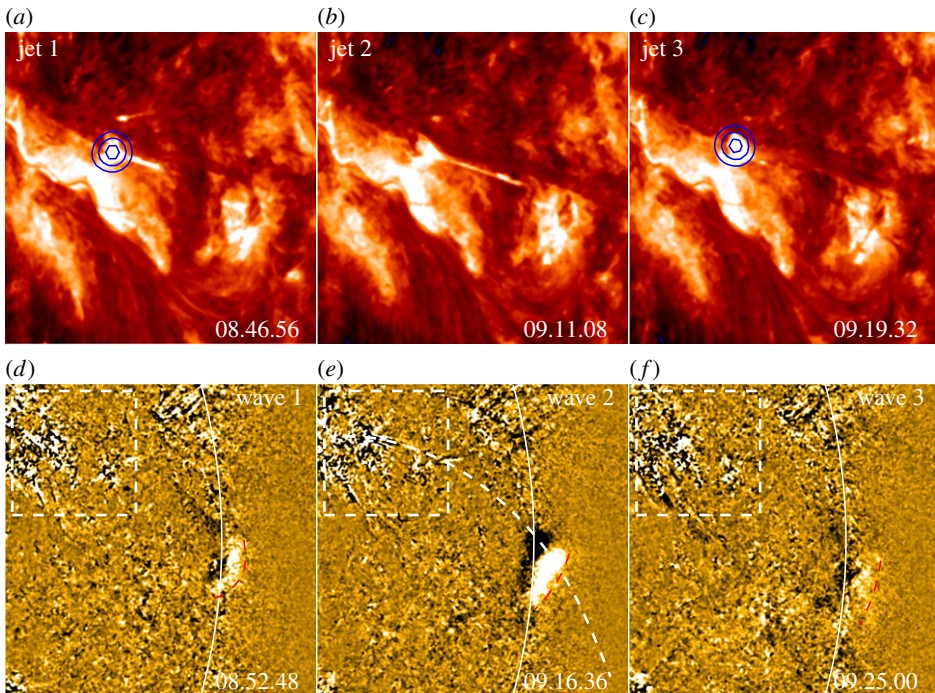

**Figure 8.** EUV waves driven by recurrent jets [94]. (*a*–*c*) show three recurrent jets with the *SDO* 304 Å images, while the (*d*–*f*) display the corresponding EUV waves with the *SDO* 171 Å running difference images. The white boxes in the 171 Å images indicate the field of view of the 304 Å images. The blue contours in (*a*) and (*c*) indicate the *RHESSI* HXR sources, while the dashed red curves in the 171 Å images indicate the EUV waves. (Online version in colour.)

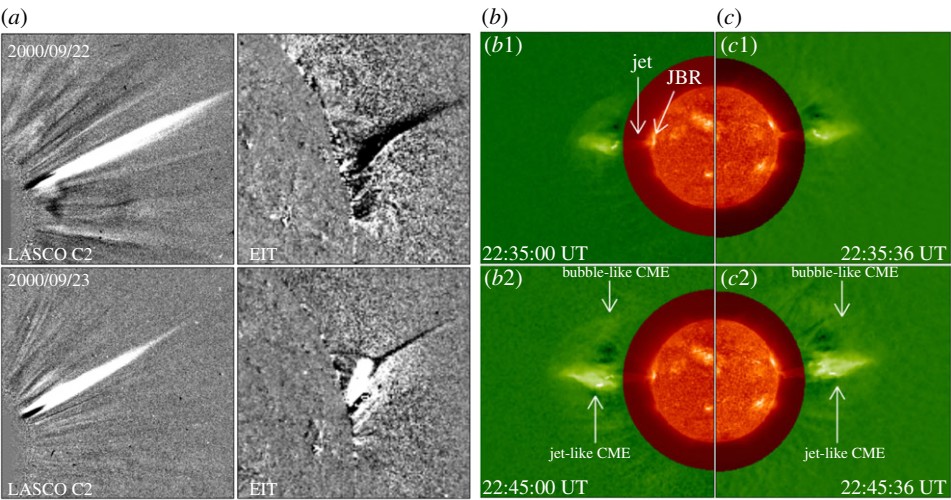

**Figure 9.** (*a*) *SOHO* LASCO/C2 and 195 Å difference images show two narrow white-light jets in the coronagraph and on the disc limb [165]. (*b,c*) *STEREO* 304 Å and the inner coronagraph difference images show a pair of simultaneous narrow and broad CMEs from two different viewing angles [239]. The bright feature in the source region and the jet are indicated by the arrows in (*b*1), while the narrow jet-like CME and the broad bubble-like CME are indicated by the arrows in (*b*2) and (*c*2), respectively. (Online version in colour.)

structures [242,243]. The on-disc progenitors of jet-like CMEs include flux emergence [244–246] and the eruption of mini-filaments [75,84,96,247,248]. Panesar *et al.* [249] studied many jets at the edge of an active region; they found that six of the homologous jets resulted in the so-called streamer-puff CMEs, and the CME-producing jets tended to be faster and longer-lasting than the non-CME-producing jets. Their observations also indicated that streamer-puff CMEs are due to the blowout of twisted streamer-base loops through magnetic reconnection.

Broad bubble-like CMEs with a typical three-part structure but on much smaller scales were found to be caused by solar jets [61], which are not simply the extension of jets into the outer corona, and their generation mechanism is possibly similar to large-scale CMEs. Jiang *et al.* [250] reported that sympathetic bubble-like CMEs can be launched through the impingement of a jet on remote interconnecting loops. Hong *et al.* [247] reported a micro-CME caused by a blowout jet that exhibited many observational characteristics as those identified in large-scale CMEs, suggesting the similarity between jet-driven micro-CMEs and large-scale CMEs [37]. Liu *et al.* [240] observed a jet-associated bubble-like CME whose bright core was evolved from the jet. Solar jets in or around active regions in association with fan–spine magnetic systems are often confined [55]. However, a few studies found that some broad CMEs are evolved from fan–spine eruptions [251]. For example, Li *et al.* [252] observed a broad CME that was caused by the eruption of a complicated fan–spine system in which a large fan–spine system hosted a small one below its fan. The event started from the eruption of a mini-filament underneath the fan of the small fan–spine system, which firstly triggered the nullpoint reconnection within the small fan–spine system; then the eruption of the small fan–spine system further triggered the nullpoint reconnection within the large fan–spine system. Here, the successful formation of CMEs from fan–spine eruptions might be the result of the weak magnetic confinement of the overlying magnetic fields or sufficient energy released during the associated flares [236].

Shen *et al.* [63] reported an interesting event in which a pair of narrow and broad CMEs were dynamically connected to a single blowout jet which showed cool and hot components. A similar event was possibly observed by Ko *et al.* [116], who detected both cool and hot components in a jet and the appearance of both a jet-like and a bubble-like CME pair in the coronagraph. However, owing to the low-resolution observations they used, the authors did not establish the physical relation between the CMEs and the jet. Shen *et al.* [63] proposed a cartoon model to interpret the generation of the cool and hot components and the formation of the paired CMEs. According to their interpretation, the hot component is the outward-moving heated plasma flow generated and accelerated by the external reconnection between the base arch and the ambient open field lines, which further evolves into the narrow jet-like CME in the outer corona. In the meantime, the external reconnection removes the confining fields of the mini-filament, which therefore leads to the rising of the mini-filament and the formation of an internal current sheet between the two legs of the confining field lines. Finally, the reconnection in the internal current sheet further results in the full eruption of the mini-filament and the formation of the broad bubble-like CME, during which the erupting filament material forms the jet's cool component. Recently, more and more observations have evidenced the appearance of paired narrow and broad CMEs in association with on-disc blowout jets [155,239,243,253,254], and the phenomenological model of Shen *et al.* [63] provides a possible explanation for these observations; however, further observational and numerical investigation is required to confirm this scenario.

## (e) Particle acceleration

Solar energetic particles (SEPs) carry important information about the particle energization inside the solar corona, as well as the properties of the acceleration volume. SEP events are divided into 'gradual' and 'impulsive' types. Gradual SEP events are long-lasting, intense, more closely correlated with CMEs and characterized by the abundances and charge states of the solar wind. Therefore, they are thought to be accelerated by CME-driven coronal/interplanetary shock waves. By contrast, impulsive SEP events are short lived, less intense, closely related to

flaring active regions, characterized by high $^3$He/$^4$He ratios and high ionization states and tightly correlated with type III radio bursts [255].

Flaring regions accompanied by solar jets are found to be the most possible candidate solar source for producing impulsive SEP events [256], since the magnetic field along which a jet emerges is open to interplanetary space, offering a clear 'escape route' for flare-accelerated particles. In radio observations, type III bursts are produced by electrons streaming along open field lines extending to interplanetary space. Many studies have indicated that type III radio bursts and SEP events are spatially and temporally associated with solar jets [69,257–262]. Wang *et al.* [263] investigated 25 $^3$He-rich events and found that their sources lie close to coronal holes and are characterized by jet-like ejections along Earth-directed open field lines. Some studies suggested that impulsive SEP events are associated with narrow jet-like CMEs [263–266]. Nitta *et al.* [267] found that the solar source regions of SEP events are often accompanied by solar jets preceded by type III radio bursts, and about 80% of events showed open field lines in or around their source regions. In addition, $^3$He-rich SEPs were also observed to be associated with helical jets [268], and the solar source regions could be small active regions near coronal holes [263,266,268], plage regions [269] and sunspots [270,271].

A type III radio burst is an important diagnostic tool for SEPs. It is a signature of propagating non-thermal electron beams in a wide range of heights of the solar atmosphere (from the low corona to the interplanetary space), and is excited at the fundamental and second harmonic of the local electron plasma frequency ($f_{pe} \approx 9\sqrt{n_e}$ kHz; here, $n_e$ is the electron number density) by the Langmuir waves generated by the electron beam instabilities [272]. Since the electron number density of the corona decreases rapidly with increasing height, a type III radio burst drifts from high (low) to low (high) frequencies reflecting the upward (downward)-moving electron beams. Chen *et al.* [260,262] derived the trajectories of electron beams in the low corona and found that each group of electron beams diverges from an extremely compact region that is located behind the erupting jet spire but above the closed arcades, coinciding with the location of the magnetic reconnection predicted in jet models.

A type III radio burst is an excellent qualitative maker of accelerated electrons and their paths, but it cannot be used to quantitatively measure the emitting electron distributions because of the nonlinear processes in its generation [272]. A complementary diagnostic tool for studying accelerated electron distributions is HXR observation, which is dominated by footpoint emission in the dense chromosphere as a result of the downward-propagating electron beams, but emission from escaping electron beams in the low-density corona is typically too faint to be observed [273]. Glesener *et al.* [274] analysed the accelerated electron distributions in a jet-associated event using simultaneous HXR and microwave data and found that the HXR time profile above 20 KeV matches that of the accompanying type III and broadband gyrosynchrotron radio emission, indicating both accelerated electrons escaping outwards along the jet path and those trapped in the flare loop. Using combined radio and HXR observations, Krucker *et al.* [275] observationally confirmed the three expectant HXR sources in solar jets as those predicted in jet models, in which two sources are at the footpoints of the post-flare loop and the other one is at the footpoint of the newly formed open field lines (figure 10).

The above studies strongly indicate that impulsive SEP events are tightly associated with solar jets, and they are mostly accelerated by the mechanism of interchange reconnection. However, the detailed acceleration mechanism of SEPs is still an unresolved question, although several possible theoretical mechanisms have been proposed to account for the acceleration of SEPs [276–278].

## (f) Coronal heating and solar wind

The problems of coronal heating and the acceleration of solar wind are two highly controversial topics in solar physics. Since energy must come from the solar interior, it is hard to understand why the coronal temperature is far hotter than the solar surface. The problem is primarily concerned with how energy is continuously transported up into the corona through non-thermal processes from the solar interior and then converted into heat within a few solar radii. In the last

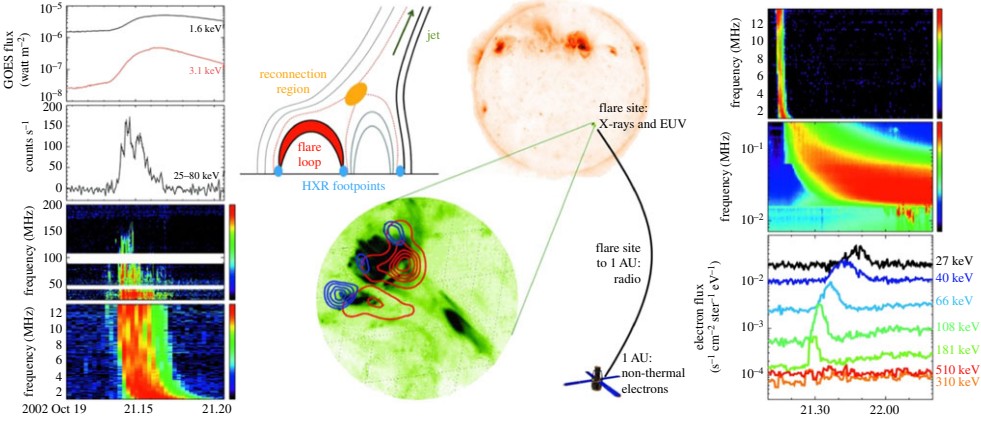

**Figure 10.** Schematic describing prompt SEP events [275]. Time-series data tracking energetic electrons from the Sun into interplanetary space are plotted on the left and right, including soft X-rays and HXRs, radio waves and non-thermal electrons seen near 1 AU. The centre shows the results of HXR and EUV observations and a schematic of the jet model. The reconnection region, the flare loop, the HXR footpoints and the jet are labelled.

half-century, many coronal heating theories have been proposed, but two theories have remained as the most likely candidates: wave heating and magnetic reconnection [279]. The solar wind is composed of charged particles including neutral atoms, positive charged ions and free electrons, which are released from the upper solar atmosphere and fill the majority of the volume of the Solar System. The solar wind has two fundamental states: slow and fast solar winds. While their compositions and temperatures are similar to the corona, their average velocities in near-Earth space are respectively about 300–500 km s$^{-1}$ and 750 km s$^{-1}$ for slow and fast solar winds [280]. Previous studies have suggested that the slow solar wind appears to originate from a region around the Sun's equatorial belt that is known as the streamer belt, while coronal holes that consist of funnel-like regions of open field lines are regarded as the solar source of the fast solar wind. However, the detailed origin and acceleration of the solar wind are still not understood and cannot be fully explained by current theory [281]. Recent observations have suggested that high-frequency but small-scale solar jets (also called spicules, fibrils and microjets) seem to be important for supplying mass and energy to power the corona and the solar wind [282].

Ultraviolet spectrum observations have revealed prevalent high-energy jets in the corona at an average speed of 400 km s$^{-1}$, whose energy and mass can satisfy the power ($6 \times 10^{27}$ erg s$^{-1}$) and mass flux ($2 \times 10^{12}$ g s$^{-1}$) requirements of the corona and solar wind if one assumes a birth rate of 24 events per second over the whole Sun [6]. Shibata *et al.* [1] proposed that chromospheric jets, which have a width (length) of 0.15–0.3 (2–5) Mm and eject at a speed of 10–20 km s$^{-1}$, may play an important role in heating coronal plasma as the nanoflare scenario [283]. The one-to-one relation between chromospheric jets and their coronal counterparts was examined in detail [284]; this showed that chromospheric plasma was propelled upwards with speeds of about 50–100 km s$^{-1}$, and with the bulk of the mass rapidly heated to a transition region temperature. A little later, plasma directly associated with these jets was heated to a coronal temperature of at least 1–2 MK, at the bottom during the initial states, and both along and towards the top of the chromospheric feature. Recently, Samanta *et al.* [9] found that enhanced coronal emission generally appeared at the top of chromospheric spicular jets, which implied that the chromospheric spicular jets channelled hot plasma into the corona. This provided a link between the magnetic activities in the lower atmosphere and coronal heating (figure 11). In addition, solar jets also excite shocks ahead of them and drive KH instability at their boundaries; the dissipation of shocks and reconnection within the vortex structures also releases energy to heat the corona [6,94,153].

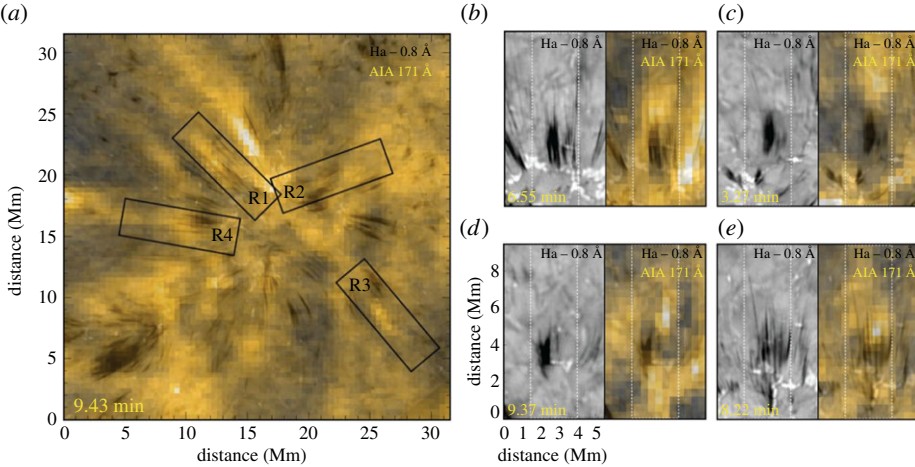

**Figure 11.** Heating effect caused by spicular activities [9]. (*a*) shows an *SDO* 171 Å image (yellow) overlaid with the Hα blue wing image (grey scale) from GST. The Hα blue wing and the same image overlain with the *SDO* 171 Å image are shown in each pair of panels in (*b*–*e*), in which the white dotted boxes correspond to the box regions (R1–R4) in (*a*). (Online version in colour.)

Alfvén waves, which propagate along magnetic field lines over large distances and transport magneto-convective energy from near the photosphere into the corona, have been invoked as a possible candidate to heat coronal plasma to millions of degrees and to accelerate the solar wind to hundreds of kilometres per second. Transverse oscillations of spicular jets were regarded as the presence or passage of Alfvén waves; the energy carried by these Alfvén waves was found to be enough to accelerate solar wind and to heat the quiet corona [285–288]. Cirtain *et al.* [7] detected two distinct speeds of solar X-ray jets, in which one is near the Alfvén speed (approx. $800 \, \mathrm{km \, s^{-1}}$) and the other is near the speed of sound (approx. $200 \, \mathrm{km \, s^{-1}}$). The authors claimed that a large number of X-ray jets with high velocities may contribute to high-speed solar wind. McIntosh *et al.* [289] found that a significant portion of the energy responsible for the transport of heated mass into the fast solar wind was provided by episodically occurring small-scale jets in the upper chromosphere and transition region. Tian *et al.* [290] found two types of Doppler shift oscillations in the corona, in which one was at the loop footpoint regions with a dominant period around 10 min while the other was associated with the upper part of the loops with a period of 3–6 min. The authors argued that the first type is quasi-periodic upflows associated with small-scale jets and plays an important role in the supply of mass and energy to the hot corona, while the second type is kink/Alfvén waves (see also De Moortel *et al.* [291] and Threlfall *et al.* [292]). Recent *IRIS* observations also reveled the prevalence of small-scale jets from the networks of the solar transition region and chromosphere [8]; they originate from small-scale bright regions and are preceded by footpoint brightenings, ejecting at a speed of $80$–$250 \, \mathrm{km \, s^{-1}}$ and accompanied by transverse waves with amplitudes of about $20 \, \mathrm{km \, s^{-1}}$. They were thought to be an intermittent but persistent source of mass and energy for solar wind.

For big jets that often reach up to a few solar radii and can be observed as white-light jets or jet-like CMEs, their contribution to solar wind often exhibits as microstreams or speed enhancements [211,293–295]. It was found that these jets are not sufficient to explain the fast solar wind [296]. Observations indicated that the motions of white-light jets are not consistent with the ballistic behaviour, and some of them even exhibit slight accelerations instead of decelerations above $3 \, R_\odot$. This suggested that the motions of white-light jets are regulated by other forces besides gravity. In addition, the bulk of almost all white-light jets travel at lower velocities, averaging around $250 \, \mathrm{km \, s^{-1}}$ at heliocentric distances of a few solar radii. These observational facts may imply that the moving jets have been incorporated into the ambient solar wind [164,241]. Yu *et al.* [297] found that fast solar polar jets show a positive correlation with high-speed responses traced into the interplanetary medium, and they contributed about 3.2% (1.6%) of the mass (energy) of solar

wind. The authors further analysed the responses in the solar wind resulting from a high-speed jet at a speed of about $1200\,\mathrm{km\,s^{-1}}$; they found a ubiquitous presence in polar coronal regions at about 100-fold mass and energy greater than the coronal response itself. This suggests that the primary acceleration of the solar wind should induce the dissipation of high-speed solar jets [298].

# 4. Physical interpretation and modelling

With the unceasing improvement of solar telescopes and numerical modelling, the physical interpretation of solar jets has achieved significant progress in recent years. In theoretical studies, the mechanism of flux emergence and the onset of instability or loss of equilibrium were investigated in detail, in which the slingshot effect, untwisting and chromospheric evaporation were considered as the possible acceleration mechanisms [33]. As more and more observational studies revealed that solar jets are caused by the eruption of mini-filaments in association with flux cancellations, new models were also developed to account for these new features. Although there are various aspects that have not yet been fully addressed, the results of the current simulations have been in reasonably good agreement with the observations, including the morphology, velocities and basic plasma properties.

## (a) Emerging-reconnection model

Heyvaerts *et al.* [299] proposed an emerging-reconnection model for explaining solar flares and surges. This mechanism was also proposed for the onset of CMEs [300]. Shibata *et al.* [105] found that many X-ray jets are associated with emerging flux regions, and started with the formation and ejection of magnetic plasmoids; they therefore proposed that the emerging-reconnection scenario could be a possible explanation for solar jets. This scenario was tested with 2D MHD simulation without considering the effect of heat conduction and radiative cooling [129,140], which showed that simultaneous hot X-ray jets and cool Hα surges are generated by magnetic reconnection between emerging fluxes and ambient pre-existing magnetic fields (figure 12). The hot jet is the secondary jet accelerated by the enhanced thermal pressure gradient behind the fast shock caused by the collision of the reconnection outflow with the ambient magnetic field, while the cool surge is formed by the cold chromospheric plasma that is carried up by the emerging flux and accelerated by the tension force of the reconnected field lines (slingshot effect). The reconnection outflow from the current sheet is composed of many plasmoids produced by tearing and coalescence instabilities, which represent miniature flux ropes in three dimensions [105,144–146]. The emerging-reconnection scenario was intensively studied in previous articles with 2D and 3D simulations by considering more realistic physical conditions, and the results could be applied to explain many characteristics of solar jets (see [33,145,301–303] for details).

Although there are many theoretical studies of the emerging-reconnection scenario in the literature, the explicit observational evidence for flux emergence directly driving jets is so far limited. In an emerging active region, Li *et al.* [163] observed 575 jets, most of which occurred at the periphery of the emerging fluxes. However, the authors did not determine the relationship between the jets and the flux emergence. In many observations, opposite polarities firstly show emergence, which is then followed by flux cancellation, and the associated jets often occurred right after the start of flux cancellation [37,46,63,69,87]. Such a magnetic flux variation pattern suggests that the trigger for the jets should be flux cancellations. One possible example of this case was presented by Cheung *et al.* [194]; however, since the jet eruption source region was very dynamic with mixed polarity, it hard to say if the jet was directly caused by flux emergence or not. Panesar *et al.* [98] argued that the reason for flux emergence not directly causing jets is because the speed of emergence is seldom, if ever, fast enough. The emerging-reconnection scenario was challenged by the discovery of blowout jets that often involve the eruption of mini-filaments or filament channels [36,62,63,69,97] in association with flux cancellations [44,66,68,247,304–308], and the cool component of blowout jets is actually the erupting filament material itself [44,63,69] rather than chromospheric material carried up by the emerging flux and accelerated by the

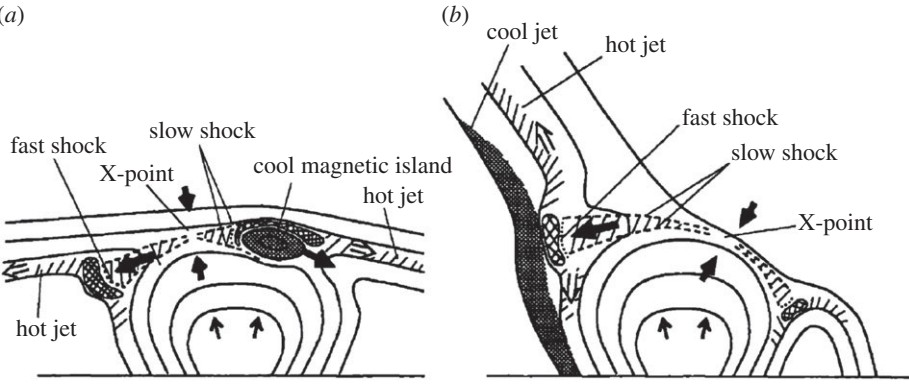

**Figure 12.** Schematic diagrams of the numerical models of two-sided-loop jets (*a*) and anemone jets (*b*), in which the arrows show the direction of movement of the magnetic field lines, and features including the X-point, slow and fast shocks, and cool and hot jets are indicated [140].

magnetic tension force as proposed in the emerging-reconnection model [129]. In a statistical study of 27 equatorial coronal hole jets, Kumar *et al.* [97] found that six jets (22%) were apparently associated with flux cancellations, while the remaining events did not show measurable flux emergence or cancellation associated with the eruption. Therefore, the ultimate trigger source of solar jets still requires further investigation. One should keep in mind that the emergence aspects of the models introduced in the following sub-sections is still subject to verification.

## (b) Embedded-bipole model

Pariat *et al.* [309] proposed the embedded-dipole model to interpret rotating solar jets. The model adopted an axisymmetrical fan–spine topology that hosts a nullpoint within the system, in which magnetic free energy is injected slowly by footpoint motions that introduce twist within the closed dome and is released rapidly by the onset of an ideal kink instability. Since reconnection is forbidden for the axisymmetrical nullpoint topology, explosive energy release via reconnection can only occur when the symmetry of the system is broken by the occurrence of kink instability until the magnetic stress builds up to a high level. The reconnection between the twisted, close and the ambient untwisted, open field lines launches a torsional Alfvén wave which compresses and accelerates the plasma along the reconnected open field lines upwardly. Eventually, an upward-ejecting helical rotating jet is generated, which has similar geometrical features, such as the inverted-Y shape, the drift of the jet axis [105], the helical structure [2,63,180] and Alfvén waves within jets [7,199]. It was found that this mechanism can efficiently release about 90% of the free energy stored in the embedded bipole topology. If a stress is constantly applied at the photospheric boundary, recurrent rotating jets can be launched [310]. In such a symmetric configuration, Rachmeler *et al.* [311] found that reconnection is fundamental for jet formation. Recently, the embedded-bipole model was subsequently extended to study the influence of magnetic field geometry [312], plasma beta [313], gravity [314] and the characteristic lengths of the spine and fan structures [315]. In addition, the possible applications of the embedded-dipole model to interpret standard and blowout jets [312] and the formation of plasmoids [146], microstreams and torsional Alfvén waves in the solar wind [314] were also explored in great detail.

## (c) Breakout jet model

The magnetic breakout model was originally proposed to interpret the initiation of large-scale CMEs, in which magnetic reconnection between the unsheared field and neighbouring flux systems decreases the amount of the overlying field and, thereby, allows the low-lying sheared

flux to break out [316]. So far, the magnetic breakout model has been confirmed by many observational studies [237,317].

Recently, high-resolution observational and statistical studies suggested that all coronal jets are probably driven by mini-filament eruptions, and they share many common characteristics with large-scale eruptions. Therefore, coronal jets are proposed to be the miniature version of large-scale eruptions [33,36,62–64,74,186,247]. In this line of thought, Wyper *et al.* [318] performed an ultra-high-resolution 3D MHD simulation to test this hypothesis, using the above-mentioned embedded-dipole scenario (figure 13). The initial magnetic configuration is a fan–spine structure, which is current-free and therefore has no filament and no free energy within it to power an eruption. Through shearing the footpoints of field lines connected to the parasitic polarity over a finite time interval, the system is energized and a twisted filament structure is generated underneath the fan structure. The confining field lines of the filament expands upwards towards the nullpoint and creates a current sheet between the confining field and the ambient open field. The (external) reconnection in this current sheet removes the confining field of the filament, allowing the filament to rise. Subsequently, (internal) reconnection starts underneath the filament, possibly enhanced by the kink or torus instability, which eventually leads to the violent eruption of the system and the formation of a rotating jet along the reconnected open field lines. The simulated jet is accelerated by torsional Alfvén waves launched when the twist in the filament begins to transfer into the ambient open field through magnetic reconnection, and the jet body is composed of hot and cool plasma flows originating from the reconnection region and the filament, respectively. In this model, the eruption is due to reconnection rather than ideal instability as proposed in the embedded bipole model [309], and the physical process is similar to the magnetic breakout model. Therefore, the authors named their model the breakout jet model, and claimed that the magnetic breakout model is a universal model for solar eruptions regardless of their scales. In subsequent studies, the authors further used their model to explain observational features of solar jets [319–321].

## (d) Data-constrained and data-driven models

To obtain realistic numerical results that are more comparable to real observations, some works managed to use multi-wavelength observations in tandem with MHD simulations to investigate the formation and evolution of solar jets. Such simulations are known as data-driven models, which use continuously time-varying solar observations as the input to reproduce solar jets. By contrast, if one use only an instantaneous cadence of observation as the input, it should be called data-constrained modelling.

Jiang *et al.* [322] simulated a jet-like eruption in a realistic and self-consistent way from its origin to onset with a data-driven MHD model; their result is well consistent with EUV observations. The authors found that the transition from the pre-eruptive to eruptive state is due to the magnetic reconnection between a stressed emerging and expanding arcade and the ambient pre-existing open field, in agreement with the physical picture described in anemone jet models [129,301]. In addition, their simulation also revealed that the non-potential magnetic flux emergence not only continuously injects magnetic free energy/helicity into the system owing to photospheric shearing motions but also stresses the field to form an intense current sheet.

Using an extrapolated non-force-free magnetic field as the initial condition, Nayak *et al.* [323] performed a data-constrained MHD simulation to study blowout jets. In their simulation, the plasma is idealized to be incompressible, thermally homogeneous and having perfect electrical conductivity. They found that the initiation of the jet is due to the magnetic reconnection near a set of two 3D magnetic nullpoints, and the jet itself evolves from a flux rope near the nullpoints through changing the flux rope's magnetic field lines from an anchored to an open topology. In addition, the generation of flare ribbons is found to be attributed to reconnections at a 3D nullpoint and a quasi-separatrix layer, consistent with a previous data-constrained simulation of circular flares [50] and the observations of confined fan–spine jets [55].

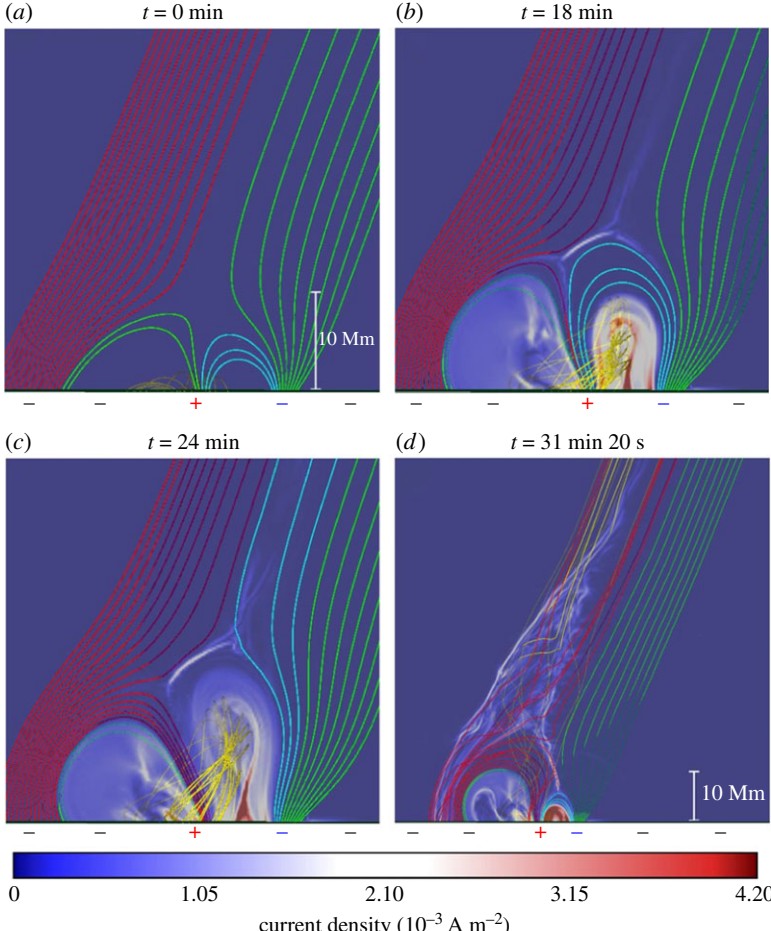

**Figure 13.** The evolution configuration of the breakout jet model [318]. Field lines with different colours represent different connectivity domains, and the positive and negative polarities are, respectively, indicated by the plus and minus symbols. The yellow curves depict the filament or flux rope formed beneath the central arcade. The current density is displayed as semi-transparent shading (colour scale), and high current density regions can be identified as thin strips beneath the filament (*d*) and the centre of the simulation domain (*b*,*c*). (Online version in colour.)

Cheung *et al.* [194] presented the data-constrained simulations of four homologous helical jets originating from a fan–spine magnetic system. Based on the extrapolated potential magnetic field, the authors used the time-dependent magneto-frictional method [324] to carry out a numerical simulation of the coronal field evolution. Their result showed that the emergence of a current-carrying magnetic field supplies the magnetic twist needed for the formation of recurrent helical jets. Since the magneto-frictional method calculates the evolution of the magnetic field through a series of quasi-static equilibria in response to photospheric footpoint motions, it can capture the response of the relaxation of a magnetic configuration to the Lorentz force, but it cannot reveal the heating process of the cool plasma by the stored magnetic energy, as well as the acceleration mechanism of the ejecting plasma. Meyer *et al.* [325] presented eight different simulations to demonstrate the structure of coronal jets in unipolar regions, in which the coronal magnetic field is evolved in time using the magneto-frictional technique. The investigated photospheric magnetic field configurations include a single parasitic polarity rotating or moving in a circular path and opposite polarity pairs involved in flyby (shearing), cancellation and emergence. Although the simulations cannot model the dynamic eruptive stage of the jets, they can be used to diagnose the

build-up of magnetic energy and the formation of the jet structures. The authors found that certain configurations and motions, such as twisting and shearing, can produce a twisted flux rope and allow a significant build-up of free energy, and that they can be viewed as the progenitors of blowout jets; other simpler configurations are more comparable to the standard jets.

## (e) Large-scale interplanetary jets

Most previous simulations were performed within small numerical domains in Cartesian geometry to study the generation mechanism and evolution process. So far, only a few publications have considered a large simulation domain extension to the interplanetary space using spherical geometry to investigate the interplanetary effects caused by solar jets.

Török et al. [326] and Lionello et al. [327] performed a 3D, viscous, resistive MHD simulation in spherical coordinates. The simulation domain covered the corona from $1\,R_\odot$ to 20, and the effects of radiative losses, thermal conduction parallel to the magnetic field and an empirical coronal heating function were all considered. The simulation adopted the flux emergence scenario to generate the jet in the low corona, in which the authors evidenced the transition of a standard jet to a blowout jet if the emergence was imposed for a long time, resembling other 3D emerging-reconnection models [145]. A white-light CME was identified 2 h after the launch of the standard jet. Several plasmoids were identified along the CME, which manifested as the episodic reconnection outflows at larger heights. It was estimated that the total energy and mass provided by the jet to the background solar wind are about 0.3–1.0% and 0.3–3.0%, respectively. In addition, the authors found that blowout jets can produce a stronger perturbation in the solar wind than standard ones.

To investigate the influence of solar jets on the solar wind, Karpen et al. [314] extended the embedded bipole model [309] by including spherical geometry, gravity and solar wind in a non-uniform, coronal hole-like ambient atmosphere. Similar to previous works, they launched a helical jet due to the resistive kink-like instability that drives fast reconnection across the closed–open separatrix; they found that the jet propagation was sustained through the outer corona, in the form of a travelling nonlinear Alfvén wavefront trailed by slower-moving plasma density enhancements that were compressed and accelerated by the wave. The authors claimed that their results agree well with observations of white-light jets, and can explain microstreams and torsional Alfvén waves detected in situ in the solar wind. Using another code that employs Alfvén wave dissipation to produce a realistic solar wind background, Szente et al. [328] studied the effects of coronal jets on the global corona and their contribution to the solar wind. A reconnection-driven blowout jet similar to that described by Pariat et al. [309] was generated, and its physical structure, dynamics and emission closely matched the observed EUV and X-ray jets. The authors found that the large-scale corona was affected significantly by the outwardly propagating torsional Alfvén waves generated by the jet (across $40°$ in latitude and out to $24\,R_\odot$). The simulation also showed that the magnetic untwisting loses most of its energy in the low corona below $2.2\,R_\odot$, but the introduced magnetic perturbation can propagate out to $24\,R_\odot$ within 3 h. Consistent with observational results [297,298], the above simulations confirmed the conjecture that coronal jets provide only a small amount of mass and energy to the solar wind.

## 5. Conclusion and prospects

High-spatio-temporal-resolution imaging and spectroscopic and stereoscopic observations covering a wide temperature range over the last several decades have significantly improved our understanding of solar jets, including various aspects such as their triggering, formation, evolution, fine structure, relationships with other solar eruptive activities, and their possible contribution to the coronal heating and acceleration of solar wind. Nowadays, we recognize that the basic energy release mechanism in solar jets is magnetic reconnection; they are triggered by photospheric magnetic activities exhibiting as flux–flux cancellations and shearing motions of opposite polarities, and are accelerated alone or in combination by possible mechanisms of

untwisting, chromospheric evaporation and slingshot effects. Observationally, solar jets can be divided into eruptive jets and confined jets, or straight anemone jets and two-sided-loop jets; they can evolve from different progenitors including satellite sunspots (or small opposite-polarity magnetic elements), mini-filaments, coronal bright points and mini-sigmoids; they can exhibit various fine structures including cool and hot components, plasmoids and KH vortex structures; and they can show interesting rotating and transverse oscillation motions. Solar jets not only provide the necessary mass and energy to the corona and solar wind, triggering other eruptive phenomena such as EUV waves, filament and loop oscillations and CMEs, but also significantly affect the interplanetary space through launching CMEs and energetic particles. Among the new knowledge that we have gained in recent years is that solar jets are often driven by mini-filament eruptions in association with photospheric magnetic flux cancellations; in addition to narrow white-light jets, broad CMEs with typical three-part structures and simultaneous paired narrow and broad CMEs are found to be dynamically associated with solar jets. These findings lead to an important conclusion that solar jets may represent the miniature version of large-scale solar eruptions, and they probably hint at a scale invariance of solar eruptions. In this sense, investigating solar jets can provide important clues to understanding complicated large-scale solar eruptions (e.g. CMEs) and currently indistinguishable small-scale transients (e.g. spicules).

Numerical modelling of solar jets has also achieved many significant advances in recent years. MHD models of solar jets have been developed from one to three dimensions with different scenarios such as the emerging-reconnection and onset of instability mechanisms, which can be applied to interpret the formation, evolution, morphology and plasma properties of standard and blowout jets in coronal holes and active regions. Recently, some numerical works have further considered the effects of heat conduction, radiative losses and background heating, and more realistic data-constrained and data-driven MHD simulations are being developed to understand solar jets. These great efforts make the obtained numerical results more morphologically and quantifiably comparable to real observations. In addition, numerical works that consider a large domain extension to the interplanetary space using spherical geometry are also being developed to aid our understanding of the interplanetary disturbances resulting from solar jets.

Despite the great advances achieved in previous observational and numerical studies, there are still many aspects of solar jets that deserve further investigation. The following is a list of some outstanding questions.

(i) Observations have shown that solar jets are tightly associated with magnetic flux cancellation, especially in mini-filament-driven jets. Nevertheless, what kind of physical process takes place during the triggering stage is still unclear. Physically, flux cancellation represents three possible processes: emergence of U-shaped loops, submergence of $\Omega$-shaped loops and reconnection in the magnetogram layer [329]. Therefore, which process and how flux cancellations trigger a solar jet need to be clarified in future observational and numerical works. In addition, although many models have been developed based on the emergence-reconnection scenario, explicit evidence for flux emergence directly driving jets is still very limited. Therefore, these models should be verified with more observational studies.

(ii) Observational studies have indicated that solar jets not only cause narrow white-light jets in the outer corona, but they can also result in broad CMEs with a typical three-part structure. Sometimes, a single mini-filament-driven jet can cause a pair of simultaneous narrow and broad CMEs [63]. Narrow white-light jets are simply an extension of solar jets into the outer corona; however, the physical relationship between solar jets and broad and simultaneous paired narrow and broad CMEs is still unclear. Although the formation of solar jets in the low corona has been intensively studied with 3D MHD simulations, there still no theoretical or simulation studies to aid our understanding of how a straight, linear solar jet can cause broad and paired narrow and broad CMEs in the outer corona. Therefore, this aspect deserves further observational and theoretical investigations, and

this kind of study could also help us to understand the similarity between small- and large-scale solar eruptions.

(iii) Although more and more observational studies have shown the similarity between small-scale solar jets and large-scale filament/CME eruptions, the possible scale invariance of solar eruptions should be further tested theoretically and observationally. It should be made a priority to check whether the current jet models can be applied to small-scale explosions such as spicules and nano-flares, which are believed to be important for coronal heating. On the other hand, it is also important to check if the current jet models are suitable for explaining complicated large-scale solar eruptions and astrophysical jets.

(iv) The contribution of solar jets to coronal heating and to the formation and acceleration of solar wind as well as the jet-associated acceleration mechanism of SEPs should be investigated in depth. There are too many assumptions and uncertainties in the existing studies on these topics.

(v) Most of the current MHD simulations only deal with idealized boundary and initial conditions using a relatively small numerical domain. Future investigations should consider more realistic data-constrained and data-driven MHD simulations, using a large simulation domain so that one can study the interplanetary disturbances caused by solar jets.

The investigation of solar jets will benefit from future ground-based large-aperture solar telescopes and advanced space missions. For example, the *Parker Solar Probe* (*PSP* [330]), launched in 2018, will observe the Sun within $9.86\,R_\odot$ by 2025, which means that it will fly through coronal structures such as solar jets and CMEs and provide *in situ* detection of physical parameters. The *Solar Orbiter* [331], launched in 2020, will operate both in and out of the ecliptic plane, imaging the polar regions of the Sun where solar jets are prominent and providing an opportunity for stereoscopic diagnosis of solar jets in combination with other telescopes on the geosynchronous orbit. The 4 m ground-based Daniel K. Inouye Solar Telescope [332], which is under construction, has obtained its first images, which can distinguish solar features as small as 30 km in size. The ultra-high-spatial-resolution observations will help us to resolve the triggering and formation problems of solar jets, as well as small spicules. The *Advanced Space-based Solar Observatory* (*ASO-S* [333]), which will be launched in 2021, will provided coronagraph, photospheric magnetic field and HXR observations for the investigation of solar jets. A combination of the measurements of magnetic field, spectroscopy, imaging and *in situ* observations provided by the above telescopes will undoubtedly result in a significant breakthrough in our understanding of the physics of solar jets and related phenomena.

Data accessibility. This article has no additional data.

Competing interests. I declare that I have no competing interests.

Funding. This work was supported by the Natural Science Foundation of China (grant nos. 11922307, 11773068, and 11633008), the Yunnan Science Foundation (2017FB006) and the West Light Foundation of the Chinese Academy of Sciences.

Acknowledgements. The author thanks the referees, who provided many valuable suggestions and comments for improving the quality of the present paper, and is grateful for the helpful discussions with Dr Y. Liu, L. Yang and J. Hong and their careful reading of the manuscript.

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
