## [Peer Review File · Proceedings. Mathematical, Physical, and Engineering Sciences]

Review History

RSPA-2020-0217.R0 (Original submission)

Review form: Referee 1

Is the manuscript an original and important contribution to its field?

Good

Is the paper of sufficient general interest?

Excellent

Is the overall quality of the paper suitable?

Excellent

Can the paper be shortened without overall detriment to the main message?

Yes

Do you think some of the material would be more appropriate as an electronic appendix?

No

Do you have any ethical concerns with this paper?

No

Recommendation?

Accept with minor revision (please list in comments)

Comments to the Author(s)

This article reviews the observed properties of solar jets, their relation to other observed phenomena and modelling efforts to explain the observations. There have been several reviews of jets in recent years, in particular a quite comprehensive one in 2016 by Raouafi et al., which covered much the same topics. However, the field is fairly fast paced and there have been new developments in both theory and observations in the last four years that are presented here. This manuscript also covers a wider scope of related phenomena than recent previous works. Therefore, my recommendation is to accept this work subject to addressing the relatively minor issues listed below. Note page numbers are those generated for the proof (out of 38).

Page 3, Line 46 – the author may wish to include the review by Innes et al. (2016):
<https://doi.org/10.1002/asna.201612428>

Page 4, Line 20 – “fan-spine topology” is introduced and is described as having a closed outer spine. Shortly after fan-spine jets are defined as jets with 3 sets of ribbons corresponding to the inner, outer spine and circular fan foot points of the spine-fan topology. I find this terminology confusing, since essentially all jets are thought to involve the spine-fan topology. In coronal holes the outer spine is simply open. Therefore, all jets could be called fan-spine jets. A more appropriate name for these events would be confined jets, or coronal loop jets. Would the author consider using a different terminology for these confined events?

Plasmoid section – In the discussion of numerical experiments that show plasmoids three studies are described. However, all the discussed simulations are two dimensional. Several other numerical studies have observed or described plasmoids in 3D. E.g.

Archontis et al. (2006) - <https://iopscience.iop.org/article/10.1086/506203>

Moreno-Insertis & Galsgaard (2013) - <https://iopscience.iop.org/article/10.1088/0004-637X/771/1/20>

Wyper et al. (2016) - <https://iopscience.iop.org/article/10.3847/0004-637X/827/1/4/meta>
 I think these should be mentioned at the end of this section.

Page 13, line 22 – “both are transient phenomena and are rooted in network boundaries”. By jets the author is referring to small chromospheric jets, not including for instance coronal jets from the satellite spots of active region which are much bigger phenomena not rooted in the network boundaries. The loose definition of jets used throughout the paper covers all kinds of jets, so this sentence essentially suggests all jets originate in the network boundaries. I suggest rewording this to be more specific.

Page 16, line 47 – “Normally, fan-spine jets are typically confined” again the confusing terminology of fan-spine jets being a sub-class of jets that are confined is used.

Page 16, line 50 – “break the confinement of the fan-spine system”. The confinement comes from the strength of the field in the overlying coronal loops rather than the spine-fan topology itself. Would a more accurate statement not be that the energy release in the eruption must be sufficient to break the confinement of the closed magnetic field?

CME section – the cartoon model of Shen et al. (2012) is described and it is claimed that narrow and broad CMEs can be “smoothly explained by the phenomenological model proposed by Shen et al.”. The cartoon as presented in Shen et al. (2012) suggests that the observed bubble-like CME is the signature of an erupting flux rope connected at both ends to the surface. And that the jet-like CME comes from reconnection ahead of this erupting structure. This would suggest the jet-like CME must be ahead of the bubble-like CME. However, I don’t think this is consistent with what is shown in Figure 10 where the jet-like CME is embedded inside the bubble-like CME. Additionally, the idea that the flux rope is still connected at both ends until it reaches the outer corona seems highly unlikely and is not consistent with several observational and numerical studies that show that the erupting flux rope gets reconnected, opening at one end (further

below). Therefore, I think there are several aspects of this model that are questionable and that the claim that all paired jet/bubble-like CMEs is explained by it is also questionable. I suggest replacing “and they can be smoothly explained by the phenomenological model proposed by Shen et al.”

With something like “the phenomenological model of Shen et al. provides a possible explanation for these observations but further observational and numerical investigation is required to confirm this scenario.”

Page 23, line 24 – “confirmed the observations and the physical picture proposed in Shen et al.” Since the Shen et al. cartoon explicitly states that the erupting flux rope is connected to the surface at both ends (to produce the bubble-like CME in their picture s discussed above) this simulation actually disagrees with the Shen et al. model as the erupting flux rope is clearly open at one end (Fig. 14(c)). However, the two reconnection regions are a common feature. Therefore, I would suggest rewording to something like: “This simulation undergoes two reconnections, similar to the physical picture proposed by Shen et al. (2012). However, in the simulation the erupting flux rope is opened at one end by the external reconnection.”

Page 22, line 32 – “using a fan-spine topology.” This suggests they started with this magnetic field, whereas it actually forms as a result of the emergence. I suggest changing to “producing a fan-spine topology”.

Page 23, line 13 – “laughed a hot”, an entertaining typo.

Page 25, line 25 – “reconnection starts underneath the filament due to kink or torus instability”. This isn’t what the authors claim in their paper. In their model the reconnection beneath the filament starts due to the upward expansion of the filament channel and stretching of the magnetic field either side of it, much in the same manner as large-scale breakout CME calculations. The torus or kink instability may be involved but they aren’t always the main drivers. I would suggest rewording to “The (external) reconnection in this current sheet removes the confining field of the filament, allowing the filament to rise. Subsequently, (internal) reconnection starts underneath the filament possibly enhanced by the kink or torus instability, which eventually leads...”

Review form: Referee 2

Is the manuscript an original and important contribution to its field?

Good

Is the paper of sufficient general interest?

Excellent

Is the overall quality of the paper suitable?

Good

Can the paper be shortened without overall detriment to the main message?

Yes

Do you think some of the material would be more appropriate as an electronic appendix?

No

Do you have any ethical concerns with this paper?

No

Recommendation?

Accept with minor revision (please list in comments)

Comments to the Author(s)

This is a very comprehensive review of the observation and modelling of solar jets, covering various topics of solar jets including their morphology/classification, general properties, precursors, fine structures, relationship with other solar phenomena, triggering/accelerating mechanism, and potential contribution to the coronal heating and solar wind acceleration. I recommend the publication of this review paper in the RSPA after taking the following minor corrections into account.

1. It turns out to me that this paper is focused on solar jets with length equivalent to or above that of macro-spicules (~ 10 Mm). For example in lines 31 to 34 in page 38, the author claimed that the basic energy mechanism in solar jets is magnetic reconnection. While it is true with big solar jets, it is not correct with spicules (or RBEs/RREs). Spicules have been suggested to be associated with several different mechanisms including p-mode, magnetic reconnection and swirling motions. In the rest of this paper, including in the morphology and classification, general properties and modelling, it is also clear that spicules were not included. I would suggest the author to state somewhere in the introduction that aspects of spicules are not included in this review.

2. I can understand that the author is not a native speaker. However, there are many typos and grammatical mistakes in this paper. I would suggest the author take some time to identify and correct these typos and grammatical mistakes, or ask a native speaker for proof-reading. Some of them are (I have to admit that I haven't identified all the typos or grammatical mistakes):

Page 1:

Line 39: '... hosts many jet ...' -> '... hosts many jets ...'

Line 43: 'anyplace' -> 'any place'

Line 45: '... solar jet are thought to be an important source ...' -> '... solar jets are thought to be one of the important sources ...'

Line 55: '... are tend' -> '...tend'

Page 2:

Line 10 and 18: 'ground' -> 'ground-based'

Line 23: '... a board waveband ...' -> '... a broad waveband'

Line 26: '... have became ...' -> '... has become ...'

Line 42: '... solar wind ...' -> '... solar wind acceleration ...'

Line 43: '... be extend to ...' -> '... be extended to ...'

Line 46: '... can refer ...' -> 'can refer to ...'

Page 3:

Line 12: '... not very reasonable ...' -> '... not ideal ...'

Page 5:

Line 13: '... X-ray spectrum ...' -> '... X-ray spectra ...'

Page 9:

Line 12: '... jetlike events ...' -> '... jet-like events ...'

Page 10:

Line 16 to 17: This sentence should be divided into 2 for better understanding.

Page 11:

Line 29 to 30: This sentence needs to be re-written.

Page 12:

Line 21: '... in interplanetary space ...' -> '... in the interplanetary space ...'

Page 13:

Line 34: 'be expect' -> 'be expected'

Line 45: 'in the unstable and even eruption of' -> 'in the instability and even the eruption of'

Line 51: 'how does jet' -> 'how jet'

Page 15:

Line 13: 'continuous driven to' -> 'continuous driver to'

Line 47 to 48: This sentence needs to be rewritten.

Page 16:

Line 48: 'are short durations' -> 'are short lived'

Page 21:

Line 37: 'by kinematically driven' -> 'by kinematically driving'

Line 49: '... models were changed by ...' I guess you wanted to say "challenged by?"

Page 22:

Line 13: 'laughed a hot, fast jet due to the ...' ???

Page 24:

Line 9: 'between the unshared field' -> 'between the unsheared field'

Page 26:

Line 22: 'cancellation, and shearing motion' -> 'cancellation, or shearing motion'

Line 54: 'how does flux cancellation trigger' -> 'how flux cancellation triggers'

3. Other minor comments:

Page 9: The author may want to add some latest statistical studies of the lifetime, velocity, length and width of macrospicules, which are also thought as an important part of solar jets, into the section "General Property", e.g., Kiss et al. ApJ, 835, 47, 2017

Page 9, Line 29 to 30: Do you mean number density or intensity?

Page 10, Line 27 to 28: Recent statistical studies have shown the number of turns could reach above 3.6, e.g., Liu et al. Frontiers, 6, 44, 2019

Page 10, Line 29 to 40: Some other papers which might be worth mentioning on this topic: Canfield et al. ApJ, 464, 1016, 1996; Liu et al. ApJ, 852, 10, 2018; Kayshap et al. A&A 616, A99, 2018

Page 12, Line 19 to 20: A review paper which might be cited: Poletto, Living Reviews in Solar Physics, 12, 7, 2015

Page 12, Line 28: Should it not be 28 jets in Raouafi et al.?

Page 18, Line 24 to 25: Please give references and where these speeds were measured (I guess it's 1AU?)

Page 19, Line 28 to 30: Some other observational research which are highly related to this topic and support the jet-wave interaction heating mechanism: De Moortel et al. ApJ, 782, 34, 2014; Threlfall et al. A&A, 556, 124, 2013

Review form: Referee 3

Is the manuscript an original and important contribution to its field?

Good

Is the paper of sufficient general interest?

Good

Is the overall quality of the paper suitable?

Good

Can the paper be shortened without overall detriment to the main message?

No

Do you think some of the material would be more appropriate as an electronic appendix?

No

Do you have any ethical concerns with this paper?

No

Recommendation?

Major revision is needed (please make suggestions in comments)

Comments to the Author(s)

Referee report for:

Title: Invited Review: Observation and Modeling of Solar Jet

Author: Y. Shen.

This review of solar jets is well researched and contains informative information. It can be considered for publication but some important points must be addressed first.

There are two major issues that have to be addressed:

One is the need to clarify what exactly is being reviewed. "Jet" is a very general word. Perhaps the most extreme example is that it is not clear whether spicules fall into the same category as coronal jets.

The second major point is that the theoretical presentation leans heavily on the emerging-flux model for coronal jets. This material has been reviewed extensively in 2016 (reference 22), and most of the papers on this topic in the submitted manuscript were covered in that earlier review. Furthermore, although the emerging-flux idea was a natural one at the time it was suggested, since that time to my knowledge there has been very limited observational support for that idea. Therefore either references to observational support for that idea should be added, or that section should be modified and reduced substantially.

Title:

The current title requires some adjustment. Either "Observation and Modeling of a Solar Jet", or "Observation and Modeling of Solar Jets" would be acceptable grammatically. In this case, the second choice is better since many jets are being discussed.

Introduction:

- The scope of this review ("jets") is too vague and open to interpretation.

An overall difficulty that I had with this section – and this same difficulty comes up several times in the manuscript – is that it is unclear what the author means by a "jet". The first five references (1-5) are given as examples, but these cover features that could be called coronal jets, photospheric jets, transition region jets, and chromospheric spicules. Are all of these the same type of object, but just viewed on different scales, temperature ranges, etc? In particular, it is unclear whether spicules and coronal jets are made the same way, and yet there is no discussion about this.

If spicules are included, then vast changes and additions are needed. For example, the text says that jet studies began in the 1940s. But spicules were seen much earlier than that (in the 1800s)! Furthermore, the text does not cite other well-known spicule studies and reviews, such as those of Jacques Beckers (1968, 1972). It does not discuss key results from Hinode, the Swedish Solar Telescope, and other recent work. Because of this, currently in no way can this be considered a review covering spicules. Adding details of spicules would make this review much more extensive. I instead suggest that the author simply state that spicules will not be covered in detail here. It is certainly fine to talk about spicules to some degree. In fact they are already discussed in a few places in the text (for example, in the mention of p-mode waves, and in discussing possible connections to the solar wind and coronal heating). This level of discussion of spicules is fine, but it should be clarified that no attempt is being made to cover the vast topic of spicules in detail here. If desired, the author might choose to say that there is a question of whether spicules

are the same as some of the other features covered, and that the question is not yet resolved.

Similarly, some statement is needed regarding whether other features are interconnected, such as whether all surges, UV/EUV macrospicules, and coronal jets are all the same type of features. It is acceptable for the author to say that answers to these questions are not yet known, and then to state how the discussion of such features is approached in this review. A couple of possible approaches are: the author might say that the review will not directly address the question of whether the different objects are the same feature (i.e., driven by the same mechanism(s)), but will instead just present observational aspects of the different features. Another possible approach might be to say that the review will discuss these objects as if they are all different emission components of the same basic physical feature (that feature being a “coronal jet” seen in EUV and X-rays), but it cannot be ruled out that some of the objects might in fact be different, distinct, physical features, i.e. made in a different way. There are other possible approaches. The important point is to make it clear what is being studied, and to make clear what the review’s approach is regarding the possible interconnectedness of even of the features.

This or (or similar) type of revision will help to clarify what is being discussed, and the context of the review.

- The first paragraph says that jets occur with a “very high occurrence frequency....” What does “very high” mean? This should either be quantified, or reworded.
- “Jets can occur at anyplace on the Sun”. This statement is too general. Do jets occur inside of sunspot umbra? Or how about inside of individual granules? It would be better to say something like jets occur in all types of solar regions, including active regions, coronal holes, etc.
- jet-like transients that show as collimated... → .jet-like transients that manifest as collimated...
- ..energy and mass releasing into the upper... → ...energy and mass releases into the upper...
Section 2(a).

- The first paragraph talks about features “based on their different observing heights”. While this phrasing is understandable, it is not accurate because it assumes a simplistic 1-d layering of the solar atmosphere, which we know not to be strictly true. Finding the appropriate wording is tricky; one possibility is to say “based on the region of the atmosphere where they originate”, or perhaps: “based on the temperature of the atmosphere in which they occur”.

- The second paragraph says: “It was found that anemone jets could be further divided into inverted-Y type and lambda type due to their different reconnection sites between emerging arches and the ambient open fields [36].”

This is not something that was “found”, since the researchers did not observe this emergence. Instead, the authors *interpreted* the results in the way the proposed theoretical flux-emergence model. The text should be rewritten in an appropriate fashion to reflect this.

- Paragraph three discusses the percentage of blowout vs standard jets based on Moore et al (2010). Table 1 of Moore et al (2013, ApJ, 769, 134) updates these percentages to be about 50% each. The text should take note of those updated percentages.

- The same paragraph implies that Sterling et al. (reference 23) doubt the necessity of the classification of blowout/standard jets. It seems however that reference 23 (and also Moore et al 2018, ApJ, 859, 3) do not express doubt about the classification. Instead, they suggest that the mechanism for the cause of the two types of jets is different from that originally proposed in Moore et al. (2010), being due to successful/failed mini-filament eruptions rather than due to

emerging flux.

Section 2(b).

- More care is required in discussing the precursors of “jets”, because this might vary widely depending on what the author means by “jet” in any particular context (see my earlier comment about clarifying the meaning of the term “jet”).

An example of this is in the second paragraph, where it says that “the most conspicuous progenitor of solar jets in the photosphere is satellite spots”. While this is appropriate for surges (and probably many active region jets), it would not apply to coronal-hole jets, and certainly not for something like spicules or the Tian et al jets!

In the case of satellite sunspots, it might be best for the author explicitly to limit the discussion to surges, and not to speak of “jets” in general. Similar physics (magnetic interactions/reconnections) might apply in the case of coronal-hole/quiet-Sun jets, but the magnetic features would not be satellite sunspots in those cases.

It might however be true that close-proximity opposite-polarity magnetic elements is a conspicuous progenitor of many jets. Sometimes this takes the form of satellite spots, e.g. for surges. And for lower-energy jet-like features (quiet Sun and coronal hole jets for example; see below in comments about Section 4(a)), it seems to often take the form of cancelling and/or shearing opposite-polarity magnetic elements.

- Section 2(c).

Cool and hot jet components were also observed concurrently by Canfield, Reardon, Leka et al. (1996). This reference is already in the manuscript (127), and so it should be added to this discussion of hot- and cool-components also.

- The report by Zhang et al. (reference 56) of the jet X-ray component occurring 2 hours after the cool component sounds questionable, given that most jets have lifetimes of 40 min or less, according to observations reported in Section 2(d)(i) by Hinode and STEREO. The Zhang et al. X-ray data were from Yohkoh, which had unsteady cadence. Because we now know from steadier-cadence observations (such as from AIA) that jets often repeat, it is very possible that the X-ray jet of Zhang et al. came from a later jet event than the one observed in chromospheric emission.

Incidentally, this might also explain the 10-hour upper limit for lifetimes of jets reported from Yohkoh observations (this upper-limit value is not explicitly mentioned in the manuscript, but it is given in reference 116). Subsequent, higher-cadence studies never (to my knowledge) reported jets of such long duration. My guess is that this very long duration is probably due to the same reasons as stated in the previous paragraph: observations of repeating jets with unsteady Yohkoh cadence.

- The text states that Panesar et al (reference 84) found the cool jet component to occur before the hot jet component, and then the next sentence says “In contrast, many other observational studies showed that the appearance of the cool component are after the hot one [by] a few minutes”. This implies that the Panesar et al. jets had a long duration (hours?) between the time of the cool and hot jets, similar to the two hours reported by Zhang et al. But the Panesar et al jets all last 15 minutes or less according to their Table 1. So the cool and hot components must have been within a few minutes of each other for their jets too. Hence, to say regarding the Panesar et al results that “In contrast, many other observational studies showed that the appearance of the cool component are after the hot one a few minutes [37,39,74,76,85–87]” is not correct. The Panesar et al results instead seem to be consistent with those other results. [Actually I do not see where Panesar et al directly address this timing question (although I may have missed it) - my

comments here are based on looking at the jet durations in their table.]

Section 2(d)(i)

- Shimojo et al (reference 116) report that “76% of jets show constant or converging shapes”. In this statement, what does “converging shape” mean? Does “shape” mean spire width? Is so, does the width get narrower as you go out from the photosphere, or does it get narrower going from the corona down to the photosphere?

- The intensity is given as $0.7-4.0 \times 10^{-9} \text{ cm}^{-3}$. But I do not understand these intensity units (cm^{-3}).

- Paragraph 2 says that Nishizuka et al. (reference 119) discuss “chromospheric jets”. As I stated earlier, it is necessary to be clear about the terms used. In this case, it is important to clarify what is meant by chromospheric jet. For example, spicules might be described as chromospheric jets too, but it appears as if that is not what Nishizuka et al. were discussing.

Section 2(d)(iii)

- In discussing reference 140, again it should be clarified what is being referred to with the term “chromospheric jets” here.

- In the discussion of the Cirtain et al (reference 2) magneoseismology results, it should be stated where the derived physical parameters are located. For example, with a temperature of only $2 \times 10^4 - 3 \times 10^4 \text{ K}$, one would not expect this to be coronal plasma.

Section 3(a).

- The text says:

“Solar jets show some common properties with plumes; both of them are transient phenomena and are rooted in network boundaries”

I find it a big stretch to say that plumes are similar to jets for the reason that they are both transient phenomena. Jets last some tens of minutes, while plumes are tens of hours. This is very different.

- A comment: I agree with the author that the claim by Raoufi et al (reference 155) that 70% of jets were followed by the formation of plumes indeed sounds very curious, and I am happy that the author expresses the need for further study of this question.

Section 3(b).

- The references for mini-filament eruptions causing jets should also include Sterling et al (reference 23). This is because, to my knowledge, they were the first to emphasize that — for a sample of many jets — not only did the erupting mini-filament cause the jet, but also that it created a miniature flare at the location from which it is erupted, and that this explains the brightening that is often observed on one side of the jet’s base (as in Fig. 1 of the manuscript). This is analogous to the flares below erupting large-scale filaments. My reading of reference 23 is that, because of these features, they argue that their observations are at odds with the emerging-flux idea (references 90, 104), which attribute this observed brightening to an external reconnection.

The Shen et al. (reference 37) paper indeed makes excellent points regarding jets, but their cartoon and the discussion is ambiguous as to what in the cartoon should be identified with the observed brightening at the side of the jet’s base:

the location below the reconnection point, or the location labeled “flare ribbons”. Also, the cartoon of that paper is not general, because not all jets make a bubble CME. Toward the end of that paper there are suggestions regarding extensions to other blowout jets, but even then this does not encompass all coronal jets (blowout and standard). That particular paper indeed is very important for our current understanding of jets: the cartoon is very good for the event it describes, and the accompanying movie showing a mini-filament eruption is the best early example of this phenomenon. Thus that paper, along with others, have been important in changing our view of the way jets work. But other key works in forming that understanding should not be downplayed.

- Regarding the Zirin (reference 157) observation. It is probably best to use Zirin’s word for the feature that caused the filament: “surge”, rather than “jet”.

Section 3(c).

- The first paragraph talks about jets being launched by p-mode waves. Once again however the author should be sure to clarify what is being discussed with the term “jets”. The p-modes have been discussed in terms of driving dynamic fibrils and spicules (references 168 and 169), and perhaps-similar features in sunspot light bridges (reference 170). Once again however, these features might be driven by different physics from other features, such as coronal jets. As I said earlier these features can be mentioned, but the review should be careful to state that these might be radically different features (from a physics standpoint) from other jets discussed in the review.

Section 4(a).

- The text says here that “Shibata et al. (reference 66) showed observational evidence of magnetic reconnection for the emerging-reconnection scenario”. But when I look at that paper, all I see regarding that point is this statement in that paper’s Section 2: “Many of the X-ray jets are associated with flares in X-ray bright points (XBPs), emerging flux regions (EFRs), or active regions”. No magnetograms however are presented in that paper. Therefore the observations presented in 66 can be regarded only as evidence *suggesting* the emerging-reconnection scenario as a possible explanation for jets. They did not “show” such evidence.

As the author later notes in this same subsection, the evidence now is much more strongly in favor of cancellation instead of emergence as the important mechanism for generating jets. I agree with this point. Moreover, I have seen very little strong evidence that flux emergence (in the absence of cancellation) is important for production of jets. If there are such cases, then it seems as if they must be very rare. In addition to the author’s own work (27,38,39) showing that cancellation is important, there are many other examples of cancellation causing jets, for example: Hong et al. (2011, ApJ, 738L, 20), Huang et al. (2012, A&A, 548, 62), Young & Muglach (2014, PASJ, 66, 12), Young & Muglach (2014, Sol Phys, 289, 3313), Panesar et al. (2016, ApJ, 832L, 7), Panesar et al. (2018, ApJ, 853, 189), Sterling et al. (2017, ApJ, 844, 28), McGalsson et al. (2019, ApJ, 822, 16).

A paper that looked at several jets and reportedly found flux cancellation in only about 20% of the cases is Kumar et al. (2019, ApJ, 873, 93). But even in that case the authors did not report flux emergence as being important for jet formation. Thus, this paper can be presented as an argument saying that the ultimate trigger source of jets still requires investigation. But so far, at least to my knowledge, there has been little observational evidence for flux emergence (in the absence of cancellation) causing jets.

Panesar et al. (2020, ApJ, 894, 104) argues that the reason for flux emergence not causing jets is because the speed of emergence is “seldom, if ever, fast enough” to do so.

Explicit evidence for emergence directly driving jets seems to be very limited. One possible example of flux emergence directly driving a jet is presented in Cheung et al. (2015, ApJ, 801, 83).

The region in which that jet occurred in was very dynamic with mixed polarity, and so although cancellation may have occurred and perhaps caused the jet, that argument has not been made. Or perhaps that region exceeded the limit discussed by Panesar et al. (2020).

If the author knows of additional reliable observational evidence for flux emergence directly driving jets in the absence of cancellation, then that evidence or the appropriate references should be presented in this review.

Summarizing this portion of the review: unless the author presents new evidence here, there is limited evidence of the importance of flux emergence in the absence of cancellation producing jets. Many of the features of the emerging-flux model, such as the cool and hot components and the sling-shot effect, also come out of the models based on mini-filament eruptions such as that of Wyper et al (reference 272). In addition, most of the emerging-flux model presented here has already been covered in the review by Raouafi et al. (reference 22). Therefore an extensive section on the emerging-flux model for coronal jets is not needed in this review. It would be better to mention the model more briefly, provide suitable references, and indicate that observations of flux emergence causing jets so far is limited. This would help encourage future researchers to focus in on trying to clarify the observational evidence for the magnetic cause of jets, rather than blindly following theoretical ideas.

- In the discussion of two-sided-loop jets in the second-to-last paragraph of Section 4(a), the author says "It should be pointed out that the formation of two-sided-loop-jets is not always caused by reconnection between [an] emerging arch and the overlying horizontal field". My question is: is there any observational evidence that such jets are *ever* caused by that mechanism? Perhaps there is, but I am not aware of it. Either the observational evidence should be presented for this, or if not, the review should clearly say that also for these jets this is an unresolved question (or that the evidence points to cancellation being the cause, if that is the case).

Section 4, other subsections.

- Several of the other models discussed in Section 4 are cast in terms of emergence too (for example, those in subsections 4(b) and 4(d)). Again, for those cases it should be made clear that the emergence aspects of these models is still subject to verification.

Section 5.

- The conclusions should be revised to reflect the above points. In particular, the first paragraph should alter the statement that strongly says that jets are triggered by new flux emergence. Similarly, the discussion aspects of the emerging-flux model in creating jets, such as the slingshot effect, should be modified. If jets are usually driven by mini-filament eruptions instead of by flux emergence, then the sling-shot effect, etc., likely plays a role in jet production, but that role has to be reconsidered in terms of updated models; for example, models that incorporate mini-filament eruptions often triggered by flux cancellation.

Decision letter (RSPA-2020-0217.R0)

07-Aug-2020

Dear Dr Shen

The Reviews Editor of Proceedings A has now received comments from referees on the above paper and would like you to revise it in accordance with their suggestions which can be found below (not including confidential reports to the Reviews Editor).

Please submit a copy of your revised paper within four weeks - if we do not hear from you within this time then it will be assumed that the paper has been withdrawn. In exceptional circumstances, extensions may be possible if agreed with the Editorial Office in advance. Please show your changes using color in your revision, and address all the points raised by the referees in your rebuttal letter.

Please note that it is the editorial policy of Proceedings A to offer authors one round of revision in which to address changes requested by referees. If the revisions are not considered satisfactory by the Reviews Editor, then the paper will be rejected, and not considered further for publication by the journal. In the event that the author chooses not to address a referee's comments, and no scientific justification is included in their cover letter for this omission, it is at the discretion of the Editor whether to continue considering the manuscript.

- Acknowledgements
- Funding statement

To revise your manuscript, log into <http://mc.manuscriptcentral.com/prsa> and enter your Author Centre, where you will find your manuscript title listed under "Manuscripts with Decisions." Under "Actions," click on "Create a Revision." Your manuscript number has been appended to denote a revision.

You will be unable to make your revisions on the originally submitted version of the manuscript. Instead, revise your manuscript and upload a new version through your Author Centre.

When submitting your revised manuscript, you will be able to respond to the comments made by the referee(s) and upload a file "Response to Referees" in "Section 6 - File Upload". Please use this to document how you have responded to the comments, and the adjustments you have made. In order to expedite the processing of the revised manuscript, please be as specific as possible in your response to the referee(s).

IMPORTANT: Your original files are available to you when you upload your revised manuscript. Please delete any unnecessary previous files before uploading your revised version.

When revising your paper please ensure that it remains under 28 pages long. Your paper has been ESTIMATED to be 38 pages.

Once again, thank you for submitting your manuscript to Proc. R. Soc. A and I look forward to receiving your revision. If you have any questions at all, please do not hesitate to get in touch.

Yours sincerely
Raminder Shergill
proceedingsa@royalsociety.org

on behalf of
Professor Michel Destrade
Reviews Editor
Proceedings A

Reviewer(s)' Comments to Author:

Referee: 1

Comments to the Author(s)

This article reviews the observed properties of solar jets, their relation to other observed phenomena and modelling efforts to explain the observations. There have been several reviews of jets in recent years, in particular a quite comprehensive one in 2016 by Raouafi et al., which covered much the same topics. However, the field is fairly fast paced and there have been new developments in both theory and observations in the last four years that are presented here. This manuscript also covers a wider scope of related phenomena than recent previous works. Therefore, my recommendation is to accept this work subject to addressing the relatively minor issues listed below. Note page numbers are those generated for the proof (out of 38).

Page 3, Line 46 – the author may wish to include the review by Innes et al. (2016):

<https://doi.org/10.1002/asna.201612428>

Page 4, Line 20 – “fan-spine topology” is introduced and is described as having a closed outer spine. Shortly after fan-spine jets are defined as jets with 3 sets of ribbons corresponding to the inner, outer spine and circular fan foot points of the spine-fan topology. I find this terminology confusing, since essentially all jets are thought to involve the spine-fan topology. In coronal holes the outer spine is simply open. Therefore, all jets could be called fan-spine jets. A more appropriate name for these events would be confined jets, or coronal loop jets. Would the author consider using a different terminology for these confined events?

Plasmoid section – In the discussion of numerical experiments that show plasmoids three studies are described. However, all the discussed simulations are two dimensional. Several other numerical studies have observed or described plasmoids in 3D. E.g.

Archontis et al. (2006) - <https://iopscience.iop.org/article/10.1086/506203>

Moreno-Insertis & Galsgaard (2013) - <https://iopscience.iop.org/article/10.1088/0004-637X/771/1/20>

Wyper et al. (2016) - <https://iopscience.iop.org/article/10.3847/0004-637X/827/1/4/meta>

I think these should be mentioned at the end of this section.

Page 13, line 22 – “both are transient phenomena and are rooted in network boundaries”. By jets the author is referring to small chromospheric jets, not including for instance coronal jets from the satellite spots of active region which are much bigger phenomena not rooted in the network boundaries. The loose definition of jets used throughout the paper covers all kinds of jets, so this sentence essentially suggests all jets originate in the network boundaries. I suggest rewording this to be more specific.

Page 16, line 47 – “Normally, fan-spine jets are typically confined” again the confusing terminology of fan-spine jets being a sub-class of jets that are confined is used.

Page 16, line 50 – “break the confinement of the fan-spine system”. The confinement comes from the strength of the field in the overlying coronal loops rather than the spine-fan topology itself. Would a more accurate statement not be that the energy release in the eruption must be sufficient to break the confinement of the closed magnetic field?

CME section – the cartoon model of Shen et al. (2012) is described and it is claimed that narrow and broad CMEs can be “smoothly explained by the phenomenological model proposed by Shen et al.”. The cartoon as presented in Shen et al. (2012) suggests that the observed bubble-like CME is the signature of an erupting flux rope connected at both ends to the surface. And that the jet-like CME comes from reconnection ahead of this erupting structure. This would suggest the jet-like CME must be ahead of the bubble-like CME. However, I don't think this is consistent with what is shown in Figure 10 where the jet-like CME is embedded inside the bubble-like CME. Additionally, the idea that the flux rope is still connected at both ends until it reaches the outer corona seems highly unlikely and is not consistent with several observational and numerical

studies that show that the erupting flux rope gets reconnected, opening at one end (further below). Therefore, I think there are several aspects of this model that are questionable and that the claim that all paired jet/bubble-like CMEs is explained by it is also questionable. I suggest replacing

“and they can be smoothly explained by the phenomenological model proposed by Shen et al.”
With something like

“the phenomenological model of Shen et al. provides a possible explanation for these observations but further observational and numerical investigation is required to confirm this scenario.”

Page 23, line 24 – “confirmed the observations and the physical picture proposed in Shen et al.”
Since the Shen et al. cartoon explicitly states that the erupting flux rope is connected to the surface at both ends (to produce the bubble-like CME in their picture s discussed above) this simulation actually disagrees with the Shen et al. model as the erupting flux rope is clearly open at one end (Fig. 14(c)). However, the two reconnection regions are a common feature. Therefore, I would suggest rewording to something like: “This simulation undergoes two reconnections, similar to the physical picture proposed by Shen et al. (2012). However, in the simulation the erupting flux rope is opened at one end by the external reconnection.”

Page 22, line 32 – “using a fan-spine topology.” This suggests they started with this magnetic field, whereas it actually forms as a result of the emergence. I suggest changing to “producing a fan-spine topology”.

Page 23, line 13 – “laughed a hot”, an entertaining typo.

Page 25, line 25 – “reconnection starts underneath the filament due to kink or torus instability”. This isn’t what the authors claim in their paper. In their model the reconnection beneath the filament starts due to the upward expansion of the filament channel and stretching of the magnetic field either side of it, much in the same manner as large-scale breakout CME calculations. The torus or kink instability may be involved but they aren’t always the main drivers. I would suggest rewording to “The (external) reconnection in this current sheet removes the confining field of the filament, allowing the filament to rise. Subsequently, (internal) reconnection starts underneath the filament possibly enhanced by the kink or torus instability, which eventually leads...”

Referee: 2

Comments to the Author(s)

This is a very comprehensive review of the observation and modelling of solar jets, covering various topics of solar jets including their morphology/classification, general properties, precursors, fine structures, relationship with other solar phenomena, triggering/accelerating mechanism, and potential contribution to the coronal heating and solar wind acceleration. I recommend the publication of this review paper in the RSPA after taking the following minor corrections into account.

1. It turns out to me that this paper is focused on solar jets with length equivalent to or above that of macro-spicules (~ 10 Mm). For example in lines 31 to 34 in page 38, the author claimed that the basic energy mechanism in solar jets is magnetic reconnection. While it is true with big solar jets, it is not correct with spicules (or RBEs/RREs). Spicules have been suggested to be associated with several different mechanisms including p-mode, magnetic reconnection and swirling motions. In the rest of this paper, including in the morphology and classification, general properties and modelling, it is also clear that spicules were not included. I would suggest the author to state somewhere in the introduction that aspects of spicules are not included in this review.

2.I can understand that the author is not a native speaker. However, there are many typos and grammatical mistakes in this paper. I would suggest the author take some time to identify and correct these typos and grammatical mistakes, or ask a native speaker for proof-reading. Some of them are (I have to admit that I haven't identified all the typos or grammatical mistakes):

Page 1:

Line 39: '... hosts many jet ...' -> '... hosts many jets ...'

Line 43: 'anyplace' -> 'any place'

Line 45: '... solar jet are thought to be an important source ...' -> '... solar jets are thought to be one of the important sources ...'

Line 55: '... are tend' -> '...tend'

Page 2:

Line 10 and 18: 'ground' -> 'ground-based'

Line 23: '... a board waveband ...' -> '... a broad waveband'

Line 26: '... have became ...' -> '... has become ...'

Line 42: '... solar wind ...' -> '... solar wind acceleration ...'

Line 43: '... be extend to ...' -> '... be extended to ...'

Line 46: '... can refer ...' -> 'can refer to ...'

Page 3:

Line 12: '... not very reasonable ...' -> '... not ideal ...'

Page 5:

Line 13: '... X-ray spectrum ...' -> '... X-ray spectra ...'

Page 9:

Line 12: '... jetlike events ...' -> '... jet-like events ...'

Page 10:

Line 16 to 17: This sentence should be divided into 2 for better understanding.

Page 11:

Line 29 to 30: This sentence needs to be re-written.

Page 12:

Line 21: '... in interplanetary space ...' -> '... in the interplanetary space ...'

Page 13:

Line 34: 'be expect' -> 'be expected'

Line 45: 'in the unstable and even eruption of' -> 'in the instability and even the eruption of'

Line 51: 'how does jet' -> 'how jet'

Page 15:

Line 13: 'continuous driven to' -> 'continuous driver to'

Line 47 to 48: This sentence needs to be rewritten.

Page 16:

Line 48: 'are short durations' -> 'are short lived'

Page 21:

Line 37: 'by kinematically driven' -> 'by kinematically driving'

Line 49: '... models were changed by ...' I guess you wanted to say "challenged by?"

Page 22:

Line 13: 'laughed a hot, fast jet due to the ...' ???

Page 24:

Line 9: 'between the unshared field' -> 'between the unsheared field'

Page 26:

Line 22: 'cancellation, and shearing motion' -> 'cancellation, or shearing motion'

Line 54: 'how does flux cancellation trigger' -> 'how flux cancellation triggers'

3.Other minor comments:

Page 9: The author may want to add some latest statistical studies of the lifetime, velocity, length and width of macrospicules, which are also thought as an important part of solar jets, into the section "General Property", e.g., Kiss et al. ApJ, 835, 47, 2017

Page 9, Line 29 to 30: Do you mean number density or intensity?

Page 10, Line 27 to 28: Recent statistical studies have shown the number of turns could reach above 3.6, e.g., Liu et al. *Frontiers*, 6, 44, 2019

Page 10, Line 29 to 40: Some other papers which might be worth mentioning on this topic:

Canfield et al. *ApJ*, 464, 1016, 1996; Liu et al. *ApJ*, 852, 10, 2018; Kayshap et al. *A&A* 616, A99, 2018

Page 12, Line 19 to 20: A review paper which might be cited: Poletto, *Living Reviews in Solar Physics*, 12, 7, 2015

Page 12, Line 28: Should it not be 28 jets in Raouafi et al.?

Page 18, Line 24 to 25: Please give references and where these speeds were measured (I guess it's 1AU?)

Page 19, Line 28 to 30: Some other observational research which are highly related to this topic and support the jet-wave interaction heating mechanism: De Moortel et al. *ApJ*, 782, 34, 2014; Threlfall et al. *A&A*, 556, 124, 2013

Referee: 3

Comments to the Author(s)

Referee report for:

Title: Invited Review: Observation and Modeling of Solar Jet

Author: Y. Shen.

This review of solar jets is well researched and contains informative information. It can be considered for publication but some important points must be addressed first.

There are two major issues that have to be addressed:

One is the need to clarify what exactly is being reviewed. "Jet" is a very general word. Perhaps the most extreme example is that it is not clear whether spicules fall into the same category as coronal jets.

The second major point is that the theoretical presentation leans heavily on the emerging-flux model for coronal jets. This material has been reviewed extensively in 2016 (reference 22), and most of the papers on this topic in the submitted manuscript were covered in that earlier review. Furthermore, although the emerging-flux idea was a natural one at the time it was suggested, since that time to my knowledge there has been very limited observational support for that idea. Therefore either references to observational support for that idea should be added, or that section should be modified and reduced substantially.

Title:

The current title requires some adjustment. Either "Observation and Modeling of a Solar Jet", or "Observation and Modeling of Solar Jets" would be acceptable grammatically. In this case, the second choice is better since many jets are being discussed.

Introduction:

- The scope of this review ("jets") is too vague and open to interpretation.

An overall difficulty that I had with this section — and this same difficulty comes up several times in the manuscript — is that it is unclear what the author means by a "jet". The first five references (1-5) are given as examples, but these cover features that could be called coronal jets, photospheric jets, transition region jets, and chromospheric spicules. Are all of these the same type of object, but just viewed on different scales, temperature ranges, etc? In particular, it is unclear whether spicules and coronal jets are made the same way, and yet there is no discussion about this.

If spicules are included, then vast changes and additions are needed. For example, the text says that jet studies began in the 1940s. But spicules were seen much earlier than that (in the 1800s)!

Furthermore, the text does not cite other well-known spicule studies and reviews, such as those of Jacques Beckers (1968, 1972). It does not discuss key results from Hinode, the Swedish Solar Telescope, and other recent work. Because of this, currently in no way can this be considered a review covering spicules.

Adding details of spicules would make this review much more extensive. I instead suggest that the author simply state that spicules will not be covered in detail here. It is certainly fine to talk about spicules to some degree. In fact they are already discussed in a few places in the text (for example, in the mention of p-mode waves, and in discussing possible connections to the solar wind and coronal heating). This level of discussion of spicules is fine, but it should be clarified that no attempt is being made to cover the vast topic of spicules in detail here. If desired, the author might choose to say that there is a question of whether spicules are the same as some of the other features covered, and that the question is not yet resolved.

Similarly, some statement is needed regarding whether other features are interconnected, such as whether all surges, UV/EUV macrospicules, and coronal jets are all the same type of features. It is acceptable for the author to say that answers to these questions are not yet known, and then to state how the discussion of such features is approached in this review. A couple of possible approaches are: the author might say that the review will not directly address the question of whether the different objects are the same feature (i.e., driven by the same mechanism(s)), but will instead just present observational aspects of the different features. Another possible approach might be to say that the review will discuss these objects as if they are all different emission components of the same basic physical feature (that feature being a “coronal jet” seen in EUV and X-rays), but it cannot be ruled out that some of the objects might in fact be different, distinct, physical features, i.e. made in a different way. There are other possible approaches. The important point is to make it clear what is being studied, and to make clear what the review’s approach is regarding the possible interconnectedness of even of the features.

This or (or similar) type of revision will help to clarify what is being discussed, and the context of the review.

- The first paragraph says that jets occur with a “very high occurrence frequency....” What does “very high” mean? This should either be quantified, or reworded.

- “Jets can occur at anyplace on the Sun”. This statement is too general.

Do jets occur inside of sunspot umbra? Or how about inside of individual granules?

It would be better to say something like jets occur in all types of solar regions, including active regions, coronal holes, etc.

- jet-like transients that show as collimated... – > ..jet-like transients that manifest as collimated...

- ..energy and mass releasing into the upper... – > ...energy and mass releases into the upper...

Section 2(a).

- The first paragraph talks about features “based on their different observing heights”. While this phrasing is understandable, it is not accurate because it assumes a simplistic 1-d layering of the solar atmosphere, which we know not to be strictly true. Finding the appropriate wording is tricky; one possibility is to say “based on the region of the atmosphere where they originate”, or perhaps: “based on the temperature of the atmosphere in which they occur”.

- The second paragraph says: “It was found that anemone jets could be further divided into inverted-Y type and lambda type due to their different reconnection sites between emerging arches and the ambient open fields [36].”

This is not something that was “found”, since the researchers did not observe this emergence. Instead, the authors *interpreted* the results in the way the proposed theoretical flux-emergence model. The text should be rewritten in an appropriate fashion to reflect this.

- Paragraph three discusses the percentage of blowout vs standard jets based on Moore et al (2010). Table 1 of Moore et al (2013, ApJ, 769, 134) updates these percentages to be about 50% each. The text should take note of those updated percentages.

- The same paragraph implies that Sterling et al. (reference 23) doubt the necessity of the classification of blowout/standard jets. It seems however that reference 23 (and also Moore et al 2018, ApJ, 859, 3) do not express doubt about the classification. Instead, they suggest that the mechanism for the cause of the two types of jets is different from that originally proposed in Moore et al. (2010), being due to successful/failed mini-filament eruptions rather than due to emerging flux.

Section 2(b).

- More care is required in discussing the precursors of “jets”, because this might vary widely depending on what the author means by “jet” in any particular context (see my earlier comment about clarifying the meaning of the term “jet”).

An example of this is in the second paragraph, where it says that “the most conspicuous progenitor of solar jets in the photosphere is satellite spots”. While this is appropriate for surges (and probably many active region jets), it would not apply to coronal-hole jets, and certainly not for something like spicules or the Tian et al jets! In the case of satellite sunspots, it might be best for the author explicitly to limit the discussion to surges, and not to speak of “jets” in general. Similar physics (magnetic interactions/reconnections) might apply in the case of coronal-hole/quiet-Sun jets, but the magnetic features would not be satellite sunspots in those cases.

It might however be true that close-proximity opposite-polarity magnetic elements is a conspicuous progenitor of many jets. Sometimes this takes the form of satellite spots, e.g. for surges. And for lower-energy jet-like features (quiet Sun and coronal hole jets for example; see below in comments about Section 4(a)), it seems to often take the form of cancelling and/or shearing opposite-polarity magnetic elements.

- Section 2(c).

Cool and hot jet components were also observed concurrently by Canfield, Reardon, Leka et al. (1996). This reference is already in the manuscript (127), and so it should be added to this discussion of hot- and cool-components also.

- The report by Zhang et al. (reference 56) of the jet X-ray component occurring 2 hours after the cool component sounds questionable, given that most jets have lifetimes of 40 min or less, according to observations reported in Section 2(d)(i) by Hinode and STEREO. The Zhang et al. X-ray data were from Yohkoh, which had unsteady cadence. Because we now know from steady-cadence observations (such as from AIA) that jets often repeat, it is very possible that the X-ray jet of Zhang et al. came from a later jet event than the one observed in chromospheric emission.

Incidentally, this might also explain the 10-hour upper limit for lifetimes of jets reported from Yohkoh observations (this upper-limit value is not explicitly mentioned in the manuscript, but it is given in reference 116). Subsequent, higher-cadence studies never (to my knowledge) reported jets of such long duration. My guess is that this very long duration is probably due to the same reasons as stated in the previous paragraph: observations of repeating jets with unsteady Yohkoh cadence.

- The text states that Panesar et al (reference 84) found the cool jet component to occur before the hot jet component, and then the next sentence says "In contrast, many other observational studies showed that the appearance of the cool component are after the hot one [by] a few minutes". This implies that the Panesar et al. jets had a long duration (hours?) between the time of the cool and hot jets, similar to the two hours reported by Zhang et al. But the Panesar et al jets all last 15 minutes or less according to their Table 1. So the cool and hot components must have been within a few minutes of each other for their jets too. Hence, to say regarding the Panesar et al results that "In contrast, many other observational studies showed that the appearance of the cool component are after the hot one a few minutes [37,39,74,76,85–87]" is not correct. The Panesar et al results instead seem to be consistent with those other results. [Actually I do not see where Panesar et al directly address this timing question (although I may have missed it) - my comments here are based on looking at the jet durations in their table.]

Section 2(d)(i)

- Shimojo et al (reference 116) report that "76% of jets show constant or converging shapes". In this statement, what does "converging shape" mean? Does "shape" mean spire width? Is so, does the width get narrower as you go out from the photosphere, or does it get narrower going from the corona down to the photosphere?

- The intensity is given as $0.7-4.0 \times 10^9 \text{ cm}^{-3}$. But I do not understand these intensity units (cm^{-3}).

- Paragraph 2 says that Nishizuka et al. (reference 119) discuss "chromospheric jets". As I stated earlier, it is necessary to be clear about the terms used. In this case, it is important to clarify what is meant by chromospheric jet. For example, spicules might be described as chromospheric jets too, but it appears as if that is not what Nishizuka et al. were discussing.

Section 2(d)(iii)

- In discussing reference 140, again it should be clarified what is being referred to with the term "chromospheric jets" here.

- In the discussion of the Cirtain et al (reference 2) magneto-seismology results, it should be stated where the derived physical parameters are located. For example, with a temperature of only $2 \times 10^4 - 3 \times 10^4 \text{ K}$, one would not expect this to be coronal plasma.

Section 3(a).

- The text says:

"Solar jets show some common properties with plumes; both of them are transient phenomena and are rooted in network boundaries"

I find it a big stretch to say that plumes are similar to jets for the reason that they are both transient phenomena. Jets last some tens of minutes, while plumes are tens of hours. This is very different.

- A comment: I agree with the author that the claim by Raoufi et al (reference 155) that 70% of jets were followed by the formation of plumes indeed sounds very curious, and I am happy that the author expresses the need for further study of this question.

Section 3(b).

- The references for mini-filament eruptions causing jets should also include Sterling et al (reference 23). This is because, to my knowledge, they were the first to emphasize that – for a

sample of many jets — not only did the erupting mini-filament cause the jet, but also that it created a miniature flare at the location from which it is erupted, and that this explains the brightening that is often observed on one side of the jet's base (as in Fig. 1 of the manuscript). This is analogous to the flares below erupting large-scale filaments. My reading of reference 23 is that, because of these features, they argue that their observations are at odds with the emerging-flux idea (references 90, 104), which attribute this observed brightening to an external reconnection.

The Shen et al. (reference 37) paper indeed makes excellent points regarding jets, but their cartoon and the discussion is ambiguous as to what in the cartoon should be identified with the observed brightening at the side of the jet's base: the location below the reconnection point, or the location labeled "flare ribbons". Also, the cartoon of that paper is not general, because not all jets make a bubble CME. Toward the end of that paper there are suggestions regarding extensions to other blowout jets, but even then this does not encompass all coronal jets (blowout and standard). That particular paper indeed is very important for our current understanding of jets: the cartoon is very good for the event it describes, and the accompanying movie showing a mini-filament eruption is the best early example of this phenomenon. Thus that paper, along with others, have been important in changing our view of the way jets work. But other key works in forming that understanding should not be downplayed.

- Regarding the Zirin (reference 157) observation. It is probably best to use Zirin's word for the feature that caused the filament: "surge", rather than "jet".

Section 3(c).

- The first paragraph talks about jets being launched by p-mode waves. Once again however the author should be sure to clarify what is being discussed with the term "jets". The p-modes have been discussed in terms of driving dynamic fibrils and spicules (references 168 and 169), and perhaps-similar features in sunspot light bridges (reference 170). Once again however, these features might be driven by different physics from other features, such as coronal jets. As I said earlier these features can be mentioned, but the review should be careful to state that these might be radically different features (from a physics standpoint) from other jets discussed in the review.

Section 4(a).

- The text says here that "Shibata et al. (reference 66) showed observational evidence of magnetic reconnection for the emerging-reconnection scenario". But when I look at that paper, all I see regarding that point is this statement in that paper's Section 2: "Many of the X-ray jets are associated with flares in X-ray bright points (XBPs), emerging flux regions (EFRs), or active regions". No magnetograms however are presented in that paper. Therefore the observations presented in 66 can be regarded only as evidence *suggesting* the emerging-reconnection scenario as a possible explanation for jets. They did not "show" such evidence.

As the author later notes in this same subsection, the evidence now is much more strongly in favor of cancellation instead of emergence as the important mechanism for generating jets. I agree with this point. Moreover, I have seen very little strong evidence that flux emergence (in the absence of cancellation) is important for production of jets. If there are such cases, then it seems as if they must be very rare.

In addition to the author's own work (27,38,39) showing that cancellation is important, there are many other examples of cancellation causing jets, for example: Hong et al. (2011, ApJ, 738L, 20), Huang et al. (2012, A&A, 548, 62), Young & Muglach (2014, PASJ, 66, 12), Young & Muglach (2014, Sol Phys, 289, 3313), Panesar et al. (2016, ApJ, 832L, 7), Panesar et al. (2018, ApJ, 853, 189), Sterling et al. (2017, ApJ, 844, 28), McGalsson et al. (2019, ApJ, 822, 16).

A paper that looked at several jets and reportedly found flux cancellation in only about 20% of the cases is Kumar et al. (2019, ApJ, 873, 93). But even in that case the authors did not report flux emergence as being important for jet formation. Thus, this paper can be presented as an argument saying that the ultimate trigger source of jets still requires investigation. But so far, at least to my knowledge, there has been little observational evidence for flux emergence (in the absence of cancellation) causing jets.

Panesar et al. (2020, ApJ, 894, 104) argues that the reason for flux emergence not causing jets is because the speed of emergence is “seldom, if ever, fast enough” to do so.

Explicit evidence for emergence directly driving jets seems to be very limited. One possible example of flux emergence directly driving a jet is presented in Cheung et al. (2015, ApJ, 801, 83). The region in which that jet occurred in was very dynamic with mixed polarity, and so although cancellation may have occurred and perhaps caused the jet, that argument has not been made. Or perhaps that region exceeded the limit discussed by Panesar et al. (2020).

If the author knows of additional reliable observational evidence for flux emergence directly driving jets in the absence of cancellation, then that evidence or the appropriate references should be presented in this review.

Summarizing this portion of the review: unless the author presents new evidence here, there is limited evidence of the importance of flux emergence in the absence of cancellation producing jets. Many of the features of the emerging-flux model, such as the cool and hot components and the sling-shot effect, also come out of the models based on mini-filament eruptions such as that of Wyper et al (reference 272). In addition, most of the emerging-flux model presented here has already been covered in the review by Raouafi et al. (reference 22). Therefore an extensive section on the emerging-flux model for coronal jets is not needed in this review. It would be better to mention the model more briefly, provide suitable references, and indicate that observations of flux emergence causing jets so far is limited. This would help encourage future researchers to focus in on trying to clarify the observational evidence for the magnetic cause of jets, rather than blindly following theoretical ideas.

- In the discussion of two-sided-loop jets in the second-to-last paragraph of Section 4(a), the author says “It should be pointed out that the formation of two-sided-loop-jets is not always caused by reconnection between [an] emerging arch and the overlying horizontal field”. My question is: is there any observational evidence that such jets are *ever* caused by that mechanism? Perhaps there is, but I am not aware of it. Either the observational evidence should be presented for this, or if not, the review should clearly say that also for these jets this is an unresolved question (or that the evidence points to cancellation being the cause, if that is the case).

Section 4, other subsections.

- Several of the other models discussed in Section 4 are cast in terms of emergence too (for example, those in subsections 4(b) and 4(d)). Again, for those cases it should be made clear that the emergence aspects of these models is still subject to verification.

Section 5.

- The conclusions should be revised to reflect the above points. In particular, the first paragraph should alter the statement that strongly says that jets are triggered by new flux emergence. Similarly, the discussion aspects of the emerging-flux model in creating jets, such as the slingshot effect, should be modified. If jets are usually driven by mini-filament eruptions instead of by flux emergence, then the sling-shot effect, etc., likely plays a role in jet production, but that role has to

be reconsidered in terms of updated models; for example, models that incorporate mini-filament eruptions often triggered by flux cancellation.

RSPA-2020-0217.R1 (Revision)

Review form: Referee 1

Is the manuscript an original and important contribution to its field?

Good

Is the paper of sufficient general interest?

Excellent

Is the overall quality of the paper suitable?

Good

Do you think some of the material would be more appropriate as an electronic appendix?

No

Do you have any ethical concerns with this paper?

No

Recommendation?

Accept with minor revision (please list in comments)

Comments to the Author(s)

I'm happy that the author has addressed my previous comments, and I'm still happy to recommend this work for publication subject to the editor's discretion based on the paper length. However, I've noted a couple inaccuracies in the revisions that should be addressed first.

On page 6, lines 6 & 7: there is a typo regarding Sterling's classification of blowout and standard jets. It should read: successful (failed) eruptions.... blowout (standard) jets.

Also on page 6, lines 12 & 13: "in which magnetic flux cancellation is the fundamental process that not only builds the core explosive field but also triggers the eruption of that field." This is a claim made by the Moore et al. paper in relation to eruptions in general, but the wording here makes the incorrect assertion that flux cancellation is a central feature of the breakout model. The breakout model actually makes no claim as to the energy storage mechanisms, and indeed all simulations of the breakout process I've seen have no flux cancellation but use surface motions to build up free energy. Moore et al. identified external reconnection before the eruption in the majority of their jets indicating breakout reconnection is a probable trigger mechanism. They also point to flux cancellation as an energy storage mechanism and possible additional trigger mechanism, although no evidence of flux cancellation is given for their events as they are near the pole. They base this on previous quiet sun events studied by Panesar et al. Breakout and flux cancellation can happen separately, so one does not necessarily indicate the presence or need of the other. To be more accurate I suggest rewording to something along the lines of:

In a subsequent paper, Moore et al. [55] further examined 15 of the 20 jets studied by Sterling et al. [34] to study the onset of the magnetic explosion in polar coronal jets; they found that a large majority of polar jets work the same way as large-scale magnetic breakout eruptions in association with energetic flares and CMEs; in which external breakout reconnection proceeds and is involved in the triggering of the eruption. Taken together with the results of Panesar et al. (2016,2017,2018) they also claim that flux cancellation is the main process whereby the energy is stored prior to eruption in all jets and CMEs, and may also be involved in the triggering process.

Review form: Referee 2

Is the manuscript an original and important contribution to its field?

Excellent

Is the paper of sufficient general interest?

Good

Is the overall quality of the paper suitable?

Excellent

Can the paper be shortened without overall detriment to the main message?

Yes

Do you think some of the material would be more appropriate as an electronic appendix?

No

Do you have any ethical concerns with this paper?

Yes

Recommendation?

Accept as is

Comments to the Author(s)

I appreciate the efforts the author made to response my previous concerns and improve the paper. I am happy to recommend it for publication in its current form.

Review form: Referee 3

Is the manuscript an original and important contribution to its field?

Good

Is the paper of sufficient general interest?

Good

Is the overall quality of the paper suitable?

Good

Can the paper be shortened without overall detriment to the main message?

Yes

Do you think some of the material would be more appropriate as an electronic appendix?

No

Do you have any ethical concerns with this paper?

No

Recommendation?

Accept with minor revision (please list in comments)

Comments to the Author(s)

Referee report for:

Title: Invited Review: Observation and Modeling of Solar Jet

Revision 1.

Author: Y. Shen.

The revised version is greatly improved over the previous version. There are however a few points that still must be addressed.

Introduction:

- The first paragraph contains this statement:

"..they are **always** accompanied by microflares, photospheric magnetic flux cancellations, and type III radio bursts...."

Here, the word "always" is too strong. Although all of these phenomena might exist for all jets, at least for some small jets they would not be observable due to sensitivity or resolution. Moreover some points remain controversial, such as flux cancellation in some cases. Thus, "often" or "frequently" are better options than "always" here.

Section 2(a).

- In the second paragraph,

"...when a parasitic magnetic polarity emerges into a preexisting magnetic field region..."

Here, "emerges" is not the correct word, because in some cases the parasitic polarity might be carried into the region by photospheric flow instead of emerging at that location (e.g., Adams et al. 2014).

- Figure 2 caption.

The caption says that the arrow in (b1), (c1) and (d1) point to a **bright point** at the location of the small bipole. In this case it might be better to avoid the term "bright point", because it can be confused with the hot brightenings often seen at the base of jets in X-rays.

That X-ray brightening might or might not match up with brightenings in H-alpha, 304, and 193. (335 or 94 might give a better idea of the X-ray brightening location.) This review does not go into detail about the X-ray bright point, but it still might be confusing since it is discussed elsewhere under the term "jet bright point" or similar. In this case just calling it a "bright patch before the ejection" as in the original Shen et al. (2012) paper is better.

Section 2(b)

- The second paragraph says: "In quiet Sun and coronal hole regions... taking the form of small sunspots...."

Generally there are no sunspots outside of active regions, so this wording should be changed.

Section 2(c)

- The last paragraph says this"

"Shen et al. [41,52,56] found that the cool component of a solar jet is directly formed by the erupting mini-filaments confined within the jet base (see Figure 2(d5)). According to their interpretation, a mini-filament is confined by a small arch surrounded by open field lines. Due to

some reasons, the arch starts to reconnect with the ambient open field lines (external reconnection), which produces the hot jet component like the generation of a standard jet. The external reconnection not only produces the hot jet but also removes the confining field lines of the mini-filament, which further results in the instability and eruption of the filament due to the internal reconnection between the two legs of the confining field lines. Therefore, the appearance of the cool component is naturally after the hot one, and their spatial relationship is adjacent to each other. So far, more and more observations indicated the appearance of cool component in solar jets, especially in blowout jets that often involve the eruption of mini-filaments [34,36,37,53,54,60,94,115].”

This phrasing focuses on the 41, 52, and 56 contributions, but other workers also contributed to these important ideas. For example, before any of the three Shen et al. references, Moore et al. (2010) also talked about jets coming from “blowout eruptions... often carrying a filament of cool ... plasma.” Other earlier works also showed similar features. Thus a better wording would be: “Several works provide evidence that the cool component of jets... erupting minifilaments [34,36,37,41,52,53,54,56,60,94,115]. According to the interpretation of Shen et al. [41,52,56]...” Section 2(d)(ii).

- First paragraph.

“...firstly reported by Xu et al. (1984)...”

I have found an earlier reference to this: Pettit, E. 1943, ApJ, 98, 6., who talks about “tornado prominences”. There may be other such references too. The author may want to double check other places where statements such as “first[ly] reported...” are used, to be sure that some earlier works are not missed. Or if uncertain, more cautious language might be used instead. Section 3(f)

- Paragraph 4:

...that the motions of white-light jets do not consistent...

—>

...that the motions of white-light jets are not consistent...

Section 4(c)

- Second paragraph:

“Recently, high-resolution observational and statistical studies suggested that all coronal jets are probably driven by mini-filament eruptions, and they share many common characteristics with large-scale eruptions [41,52,56]. Therefore, coronal jets are proposed to be the miniature version of large-scale eruptions [34,35,53].

In this line of thought, Wyper et al. [285] performed an ultrahigh-resolution 3D MHD simulation to test this hypothesis....”

This again overstates the contributions of references 41, 52, and 56 compared to those of other researchers. None of the three papers 41, 52, 56 were statistical studies.

Moreover, 41 was published in 2019 (after Wyper et al.), and 56 was the same year as Wyper et al. (both 2017). Therefore Wyper et al. does not seem to be dependent upon those papers. Thus this discussion should be reworded in a more balanced and appropriate fashion.

Decision letter (RSPA-2020-0217.R1)

05-Jan-2021

Dear Dr Shen,

On behalf of the Reviews Editor, I am pleased to inform you that your Manuscript RSPA-2020-0217.R1 entitled "Observation and Modeling of Solar Jets" has been accepted for publication subject to minor revisions in Proceedings A. Please find the referees' comments below.

The reviewers have recommended publication, but also suggest some minor revisions to your manuscript. Therefore, I invite you to respond to the reviewers' comments and revise your manuscript. Please note that we have a strict upper limit of 40 pages for review papers. Please endeavour to incorporate any revisions while keeping the paper within journal limits. Your paper has been ESTIMATED to be 38 pages.

It is a condition of publication that you submit the revised version of your manuscript within 7 days. If you do not think you will be able to meet this date please let me know in advance of the due date.

To revise your manuscript, log into <https://mc.manuscriptcentral.com/prsa> and enter your Author Centre, where you will find your manuscript title listed under "Manuscripts with Decisions." Under "Actions," click on "Create a Revision." Your manuscript number has been appended to denote a revision.

You will be unable to make your revisions on the originally submitted version of the manuscript. Instead, revise your manuscript and upload a new version through your Author Centre.

When submitting your revised manuscript, you will be able to respond to the comments made by the referee(s) and upload a file "Response to Referees" in Step 1: "View and Respond to Decision Letter". You can use this to document any changes you make to the original manuscript. In order to expedite the processing of the revised manuscript, please be as specific as possible in your response to the referee(s).

IMPORTANT: Your original files are available to you when you upload your revised manuscript. Please delete any redundant files before completing the submission process.

In addition to addressing all of the reviewers' and editor's comments, your revised manuscript **MUST** contain the following sections before the reference list (for any heading that does not apply to your work, please include a comment to this effect):

- Acknowledgements
- Funding statement

See <https://royalsociety.org/journals/authors/author-guidelines/> for further details.

When uploading your revised files, please make sure that you include the following as we cannot proceed without these:

1) A text file of the manuscript (doc, txt, rtf or tex), including the references, tables (including captions) and figure captions. Please remove any tracked changes from the text before submission. PDF files are not an accepted format for the "Main Document".

2) A separate electronic file of each figure (tif, eps or print-quality pdf preferred). The format should be produced directly from original creation package, or original software format.

3) Electronic Supplementary Material (ESM): all supplementary materials accompanying an accepted article will be treated as in their final form. Note that the Royal Society will not edit or typeset supplementary material and it will be hosted as provided. Please ensure that the supplementary material includes the paper details where possible (authors, article title, journal name). Supplementary files will be published alongside the paper on the journal website and posted on the online figshare repository (<https://figshare.com>). The heading and legend provided for each supplementary file during the submission process will be used to create the figshare page, so please ensure these are accurate and informative so that your files can be found in searches. Files on figshare will be made available approximately one week before the accompanying article so that the supplementary material can be attributed a unique DOI. Alternatively you may upload a zip folder containing all source files for your manuscript as described above with a PDF as your "Main Document". This should be the full paper as it appears when compiled from the individual files supplied in the zip folder.

Article Funder

Please ensure you fill in the Article Funder question on page 2 to ensure the correct data is collected for FundRef (<http://www.crossref.org/fundref/>).

Media summary

Please ensure you include a short non-technical summary (up to 100 words) of the key findings/importance of your paper. This will be used for to promote your work and marketing purposes (e.g. press releases). The summary should be prepared using the following guidelines:

- *Write simple English: this is intended for the general public. Please explain any essential technical terms in a short and simple manner.
- *Describe (a) the study (b) its key findings and (c) its implications.
- *State why this work is newsworthy, be concise and do not overstate (true 'breakthroughs' are a rarity).
- *Ensure that you include valid contact details for the lead author (institutional address, email address, telephone number).

Cover images

We welcome submissions of images for possible use on the cover of Proceedings A. Images should be square in dimension and please ensure that you obtain all relevant copyright permissions before submitting the image to us. If you would like to submit an image for consideration please send your image to proceedingsa@royalsociety.org

Once again, thank you for submitting your manuscript to Proceedings A and I look forward to receiving your revision. If you have any questions at all, please do not hesitate to get in touch.

Best wishes
Raminder Shergill
proceedingsa@royalsociety.org
Proceedings A

on behalf of
Professor Michel Destrade
Reviews Editor
Proceedings A

Reviewer(s)' Comments to Author:

Referee: 3

Comments to the Author(s)

Referee report for:

Title: Invited Review: Observation and Modeling of Solar Jet

Revision 1.

Author: Y. Shen.

The revised version is greatly improved over the previous version. There are however a few points that still must be addressed.

Introduction:

- The first paragraph contains this statement:

"..they are **always** accompanied by microflares, photospheric magnetic flux cancellations, and type III radio bursts...."

Here, the word "always" is too strong. Although all of these phenomena might exist for all jets, at least for some small jets they would not be observable due to sensitivity or resolution.

Moreover some points remain controversial, such as flux cancellation in some cases. Thus, "often" or "frequently" are better options than "always" here.

Section 2(a).

- In the second paragraph,

"...when a parasitic magnetic polarity emerges into a preexisting magnetic field region..."

Here, "emerges" is not the correct word, because in some cases the parasitic polarity might be carried into the region by photospheric flow instead of emerging at that location (e.g., Adams et al. 2014).

- Figure 2 caption.

The caption says that the arrow in (b1), (c1) and (d1) point to a **bright point** at the location of the small bipole. In this case it might be better to avoid the term "bright point", because it can be confused with the hot brightenings often seen at the base of jets in X-rays.

That X-ray brightening might or might not match up with brightenings in H-alpha, 304, and 193. (335 or 94 might give a better idea of the X-ray brightening location.) This review does not go into detail about the X-ray bright point, but it still might be confusing since it is discussed elsewhere under the term "jet bright point" or similar. In this case just calling it a "bright patch before the ejection" as in the original Shen et al. (2012) paper is better.

Section 2(b)

- The second paragraph says: "In quiet Sun and coronal hole regions... taking the form of small sunspots...."

Generally there are no sunspots outside of active regions, so this wording should be changed.

Section 2(c)

- The last paragraph says this"

"Shen et al. [41,52,56] found that the cool component of a solar jet is directly formed by the erupting mini-filaments confined within the jet base (see Figure 2(d5)). According to their interpretation, a mini-filament is confined by a small arch surrounded by open field lines. Due to some reasons, the arch starts to reconnect with the ambient open field lines (external reconnection), which produces the hot jet component like the generation of a standard jet. The external reconnection not only produces the hot jet but also removes the confining field lines of the mini-filament, which further results in the instability and eruption of the filament due to the internal reconnection between the two legs of the confining field lines. Therefore, the appearance of the cool component is naturally after the hot one, and their spatial relationship is adjacent to each other. So far, more and more observations indicated the appearance of cool component in solar jets, especially in blowout jets that often involve the eruption of mini-filaments [34,36,37,53,54,60,94,115]"

This phrasing focuses on the 41, 52, and 56 contributions, but other workers also contributed to these important ideas. For example, before any of the three Shen et al. references, Moore et al. (2010) also talked about jets coming from "blowout eruptions... often carrying a filament of cool ... plasma." Other earlier works also showed similar features. Thus a better wording would be:

"Several works provide evidence that the cool component of jets... erupting minifilaments [34,36,37,41,52,53,54,56,60,94,115]. According to the interpretation of Shen et al. [41,52,56]...."

Section 2(d)(ii).

- First paragraph.

"...firstly reported by Xu et al. (1984)...."

I have found an earlier reference to this: Pettit, E. 1943, ApJ, 98, 6., who talks about "tornado prominences". There may be other such references too. The author may want to double check other places where statements such as "first[ly] reported...." are used, to be sure that some earlier works are not missed. Or if uncertain, more cautious language might be used instead.

Section 3(f)

- Paragraph 4:

...that the motions of white-light jets do not consistent...

– >

...that the motions of white-light jets are not consistent...

Section 4(c)

- Second paragraph:

"Recently, high-resolution observational and statistical studies suggested that all coronal jets are probably driven by mini-filament eruptions, and they share many common characteristics with large-scale eruptions [41,52,56]. Therefore, coronal jets are proposed to be the miniature version of large-scale eruptions [34,35,53].

In this line of thought, Wyper et al. [285] performed an ultrahigh-resolution 3D MHD simulation to test this hypothesis...."

This again overstates the contributions of references 41, 52, and 56 compared to those of other researchers. None of the three papers 41, 52, 56 were statistical studies. Moreover, 41 was published in 2019 (after Wyper et al.), and 56 was the same year as Wyper et al. (both 2017).

Therefore Wyper et al. does not seem to be dependent upon those papers. Thus this discussion should be reworded in a more balanced and appropriate fashion.

Referee: 2

Comments to the Author(s)

I appreciate the efforts the author made to response my previous concerns and improve the paper. I am happy to recommend it for publication in its current form.

Referee: 1

Comments to the Author(s)

I'm happy that the author has addressed my previous comments, and I'm still happy to recommend this work for publication subject to the editor's discretion based on the paper length.

However, I've noted a couple inaccuracies in the revisions that should be addressed first.

On page 6, lines 6 & 7: there is a typo regarding Sterling's classification of blowout and standard jets. It should read: successful (failed) eruptions.... blowout (standard) jets.

Also on page 6, lines 12 & 13: "in which magnetic flux cancellation is the fundamental process that not only builds the core explosive field but also triggers the eruption of that field."

This is a claim made by the Moore et al. paper in relation to eruptions in general, but the wording here makes the incorrect assertion that flux cancellation is a central feature of the breakout model. The breakout model actually makes no claim as to the energy storage mechanisms, and indeed all simulations of the breakout process I've seen have no flux cancellation but use surface motions to build up free energy. Moore et al. identified external reconnection before the eruption in the majority of their jets indicating breakout reconnection is a probable trigger mechanism. They also point to flux cancellation as an energy storage mechanism and possible additional trigger mechanism, although no evidence of flux cancellation is given for their events as they are near the pole. They base this on previous quiet sun events studied by Panesar et al. Breakout and flux cancellation can happen separately, so one does not necessarily indicate the presence or need of the other. To be more accurate I suggest rewording to something along the lines of:

In a subsequent paper, Moore et al. [55] further examined 15 of the 20 jets studied by Sterling et al. [34] to study the onset of the magnetic explosion in polar coronal jets; they found that a large majority of polar jets work the same way as large-scale magnetic breakout eruptions in association with energetic flares and CMEs; in which external breakout reconnection proceeds and is involved in the triggering of the eruption. Taken together with the results of Panesar et al. (2016,2017,2018) they also claim that flux cancellation is the main process whereby the energy is stored prior to eruption in all jets and CMEs, and may also be involved in the triggering process.

Decision letter (RSPA-2020-0217.R2)

12-Jan-2021

Dear Dr Shen

On behalf of the Editor, I am pleased to inform you that your manuscript entitled "Observation and Modeling of Solar Jets" has been accepted in its final form for publication in Proceedings A.

Our Production Office will be in contact with you in due course. You can expect to receive a proof of your article soon. Please contact the office to let us know if you are likely to be away from e-mail in the near future. If you do not notify us and comments are not received within 5 days of sending the proof, we may publish the paper as it stands.

Under the terms of our licence to publish you may post the author generated postprint (ie. your accepted version not the final typeset version) of your manuscript at any time and this can be made freely available. Postprints can be deposited on a personal or institutional website, or a recognised server/repository. Please note however, that the reporting of postprints is subject to a media embargo, and that the status the manuscript should be made clear. Upon publication of the definitive version on the publisher's site, full details and a link should be added.

You can cite the article in advance of publication using its DOI. The DOI will take the form: 10.1098/rspa.XXXX.YYYY, where XXXX and YYYY are the last 8 digits of your manuscript number (eg. if your manuscript number is RSPA-2017-1234 the DOI would be 10.1098/rspa.2017.1234).

For tips on promoting your accepted paper see our blog post:
<https://royalsociety.org/blog/2020/07/promoting-your-latest-paper-and-tracking-your-results/>

Thank you for your submission. On behalf of the Editors of the journal, we look forward to your continued contributions to the Journal.

Best wishes
Raminder Shergill,
Proceedings A Editorial Office
proceedingsa@royalsociety.org